# Episodic long-term memory formation during slow-wave sleep

**Flavio J Schmidig[1]\*[†], Simon Ruch[1,2], Katharina Henke[1]**

[1]Institute of Psychology, University of Bern, Bern, Switzerland; [2]Faculty of Psychology, UniDistance Suisse, Brig, Switzerland

**Abstract** We are unresponsive during slow-wave sleep but continue monitoring external events for survival. Our brain wakens us when danger is imminent. If events are non-threatening, our brain might store them for later consideration to improve decision-making. To test this hypothesis, we examined whether novel vocabulary consisting of simultaneously played pseudowords and translation words are encoded/stored during sleep, and which neural-electrical events facilitate encoding/storage. An algorithm for brain-state-dependent stimulation selectively targeted word pairs to slow-wave peaks or troughs. Retrieval tests were given 12 and 36 hr later. These tests required decisions regarding the semantic category of previously sleep-played pseudowords. The sleep-played vocabulary influenced awake decision-making 36 hr later, if targeted to troughs. The words' linguistic processing raised neural complexity. The words' semantic-associative encoding was supported by increased theta power during the ensuing peak. Fast-spindle power ramped up during a second peak likely aiding consolidation. Hence, new vocabulary played during slow-wave sleep was stored and influenced decision-making days later.

**\*For correspondence:**
flavio.schmidig@unibe.ch

[†]Lead contact

**Competing interest:** The authors declare that no competing interests exist.

## eLife assessment

This manuscript supports the intriguing idea that some aspects of novel learning can occur during sleep and outside of awareness. The authors provide **solid** evidence that presenting participants with novel words and their translations during sleep, especially during slow oscillation troughs, leads to the ability to categorize the semantic meaning of those words during awake testing 36 hours later. These findings represent a **valuable** contribution to the literature on unconscious processing and learning during sleep, although the claim that the results reflect episodic memory formation, in particular, deviates from the typical use of this term in the literature.

## Introduction

Falling asleep is accompanied by a gradual loss of consciousness of the external environment. Although the thalamic gating hypothesis had claimed a blockade of sensory information at the thalamic level (*McCormick and Bal, 1994*), we now know that sleepers still monitor the environment for their safety and survival (*Ai et al., 2018*; *Andrillon et al., 2016*; *Andrillon et al., 2017*; *Arzi et al., 2012*; *Blume et al., 2017*; *Koroma et al., 2022*; *Ruch et al., 2014*; *Ruch and Henke, 2020*; *Züst et al., 2019*). The sleeping brain decides whether an external event can be disregarded, requires immediate awakening or should be stored for later consideration in the waking state (*Ameen et al., 2022*; *Blume et al., 2018*; *Formby, 1967*; *Holeckova et al., 2006*; *Moyne et al., 2022*; *Oswald et al., 1960*; *Türker et al., 2023*). Although the processing of external information during sleep versus wakefulness is reduced (*Andrillon et al., 2016*), sleep still allows for the detection of semantic incongruity (*Bastuji and García-Larrea, 1999*; *Ibáñez et al., 2006*) and the detection of rule violations (*Ruby et al., 2008*; *Strauss et al., 2015*). Furthermore, there is evidence that the sleeping brain cannot only process

information, but can store new information, ranging from tone-odor to word-word associations (*Ai et al., 2018*; *Andrillon and Kouider, 2016*; *Arzi et al., 2012*; *Ataei et al., 2023*; *de Lavilléon et al., 2015*; *Koroma et al., 2022*; *Züst et al., 2019*). However, it remains unclear whether and under what circumstances the most sophisticated form of human learning, namely episodic memory formation, can proceed during deep sleep (*Ruch and Henke, 2020*).

The term episodic memory refers to the recollection of personally experienced episodes (*Tulving, 2002*). Episodic memory formation depends on hippocampal-neocortical interactions (*Cohen and Eichenbaum, 1993*; *Henke, 2010*). Because episodic memory belongs to declarative/explicit memory, episodic memory was long associated with wakefulness and conscious awareness of events (*Gabrieli, 1998*; *Moscovitch, 2008*; *Schacter, 1998*; *Squire and Dede, 2015*; *Tulving, 2002*). In the meantime, counterevidence suggests that hippocampal-assisted episodic memory formation may also proceed without conscious awareness of the learning material (*Duss et al., 2014*; *Reber et al., 2012*; *Schneider et al., 2021*; *Züst et al., 2019*). When applying such tasks, unconscious hippocampus-assisted episodic encoding of subliminal (invisible) words was revealed (*Duss et al., 2014*; *Reber et al., 2012*).

Thus, in the current study, we define episodic memory based on its key computational properties, which are the rapid formation of new associations calling on the hippocampus' ability for one-shot relational binding, and the flexible retrieval of these associations in novel contexts. We used this definition to explore episodic verbal learning during the unconsciousness of deep sleep. We applied a sleep-learning and awake-retrieval task that requires rapid semantic associative encoding of pseudowords and German translation words, memory storage of the formed associations over days, and a cued associative retrieval that requires a flexible representation of the sleep-encoded pseudoword-word associations. As proposed and demonstrated by previous work (*Cohen and Eichenbaum, 1993*; *Henke, 2010*; *O'Reilly et al., 2014*; *O'Reilly and Rudy, 2000*), these task-enforced demands call upon the episodic memory system, according to its computational definition.

Although slow-wave sleep (deep sleep) is characterized by an average neurochemical milieu and neural functional connectivity that does not favour episodic memory formation, slow-wave sleep is not a unitary state. In fact, neuronal activity and excitability waxes and wanes during slow-wave sleep, thereby generating the eponymous electroencephalographic slow-waves that oscillate at 1 Hz and are characterised by slow-wave peaks and troughs. A peak and a trough each last around 500 ms (*Berry et al., 2015*). Because peaks of slow-waves are associated with high neural excitability and wake-like network characteristics, peaks might provide the necessary plasticity mechanisms for episodic memory formation (*Cox et al., 2014a*; *Diba and Buzsáki, 2007*). *Züst et al., 2019* reported successful paired-associate learning in humans during peaks of slow-waves recorded in a mid-day nap with the retrieval of the sleep-learned associations tested after awakening on the same day. Because peaks with their depolarized neural states are also the time windows when memories formed during the previous days are replayed, strengthened, and consolidated (*Göldi et al., 2019*; *Mölle et al., 2002*; *Mölle et al., 2011*; *Muehlroth et al., 2019*; *Staresina et al., 2015*), high-jacking peaks for de novo learning might impair ongoing memory consolidation. Troughs on the other hand are characterized by neural silence (*Cox et al., 2014a*; *Destexhe et al., 2007*; *Schabus et al., 2012*). One speculative argument in favour of troughs might be the relative absence of ongoing consolidation processes. This absence of ongoing endogenous functional activity might rise the troughs' sensory receptiveness to external events, provided that at least local neural processing remains possible during troughs (*Destexhe et al., 2007*; *Vyazovskiy and Harris, 2013*).

Hence, both peaks and troughs possess characteristics that might support sensory processing and learning, but which of these brains states provides optimal learning conditions is unknown. Furthermore, recent results suggest that the high background firing rate during the peak state favors synaptic down-scaling/depression rather than potentiation (*Bartram et al., 2017*; *Yoshida and Toyoizumi, 2023*). Hence, the trough rather than the peak state might be beneficial for de-novo memory formation during deep sleep.

We leveraged peaks and troughs of slow-waves for the linguistic processing and the ensuing paired-associate encoding of word pairs. Here, we applied slow wave phase-targeted, brain-state-dependent stimulation for de novo memory formation, instead of memory reactivation (*Ngo and Staresina, 2022*) or the modification of slow waves (*Navarrete et al., 2020*). To this aim, we simultaneously played pseudowords and translation words during either troughs or peaks (Trough/Peak condition) of

frontal slow-waves using an electroencephalography (EEG)-based brain-state-dependent stimulation algorithm of our own devising (*Ruch et al., 2022*). According to *Züst et al., 2019*, we hypothesized that the critical process of memory formation, namely paired-associate semantic encoding, is bound to peaks because only peaks provide the necessary conditions for effective hippocampal-neocortical crosstalk. Regarding the optimal state for the words' initial psycholinguistic analysis leading up to their relational encoding, we had no directed hypothesis. We further anticipated that associations formed through hippocampal-neocortical interactions would last for hours or days thanks to immediate hippocampal long-term potentiation and an immediate hippocampally triggered replay (*Frankland et al., 2001*; *Goto et al., 2021*; *Takeuchi et al., 2014*; *Tsien et al., 1996*). This hypothesis was significantly inspired by the finding of subliminally formed unconscious episodic memories lasting over 10 hr and increasing their influence on human decision-making over this time (*Pacozzi et al., 2022*).

We played 27 pairs of pseudowords (e.g. aryl) and 27 translation words (e.g. bird; nouns) in the experimental condition (EC) and pseudowords alone (e.g. egref) in the control condition (CC) during either troughs or peaks of slow-waves (*Figure 1*). Peak/Trough-targeting was manipulated between participants with 15 participants per group. Each group was played words from the experimental condition and the control condition during sleep within the same night. Word pairs (EC) and pseudowords (CC) were played four times in succession to increase the probability that the words pass the thalamic gate. The test of whether semantic associations were formed between pseudowords and translation words during deep sleep followed 12 hr and again 36 hr later in the waking state. This retrieval test required participants to assign earlier sleep-played pseudowords to one of three semantic categories: animals, tools, places. Each sleep-played translation word was a noun that belongs to one of these superordinate categories (counterbalanced between participants: nine animals, nine tools, nine places). As participants were in deep sleep, while the vocabulary was being played, participants could not consciously remember the word pairs at test. Therefore, we encouraged them to decide intuitively about category assignments. This task triggers an unconscious associative retrieval by cueing a memory reactivation solely by the sound of the pseudoword alone. Once the meaning of the pseudoword-associated translation word was reactivated in memory, this meaning needed to be converted to the appropriate superordinate semantic category. If the number of correct category assignments exceeded chance performance (33%), we attributed this excess to successful sleep-learning. Importantly, we provided no feedback regarding category assignments to leave assignment accuracy at the 36 hr retrieval uninfluenced by the preceding 12 hr retrieval.

## Results

Thirty healthy male and female volunteers were acoustically stimulated during their slow-wave night-sleep with an average of 23.98 (*SD* 3.84) pairs of pseudowords and translation words. We targeted these word pairs either to slow-wave peaks or to troughs using our own slow wave phase-targeted, brain-state-dependent stimulation algorithm (TOPOSO, *Ruch et al., 2022*). Pseudowords and translation words were simultaneously played into the right ear and left ear, respectively (*Aarons, 1990*; *Kimura, 1961*). Their presentation lasted on average 540 ms (SEM = 0.011 ms). Word pairs' sound onset was ~100 ms before the local maxima of the targeted slow-wave phase and largely fit into this half-wave (duration of slow-wave peaks/troughs: ~500 ms, duration of word pair presentation: 540 ms, overlap: 350 ms). Each word pair was presented four times in succession to enhance the odds of their successful processing. Adding over the four repetitions, a mean of 95.33 (SD = 15.41) word pairs was played per participant.

### Targeting vocabulary to troughs provided for successful sleep-learning and long-term storage

We computed a 2x2 ANOVA with the between-subjects factor Peak- versus Trough-targeting and the within-subjects factor Encoding-Test Delay (12 hr versus 36 hr). The dependent variable was retrieval accuracy expressed as the difference between the percentages of correctly retrieved associations minus the percentage of the theoretical chance level performance (33.33 %).

Over all conditions, associative retrieval performance exceeded chance performance by 2.67% (SD = 8.11), which just failed statistical significance ($F_{Intercept}(1,28) = 3.490$, p=0.077). Retrieval performance over both encoding-test delays was significantly better when word pairs were targeted to Troughs

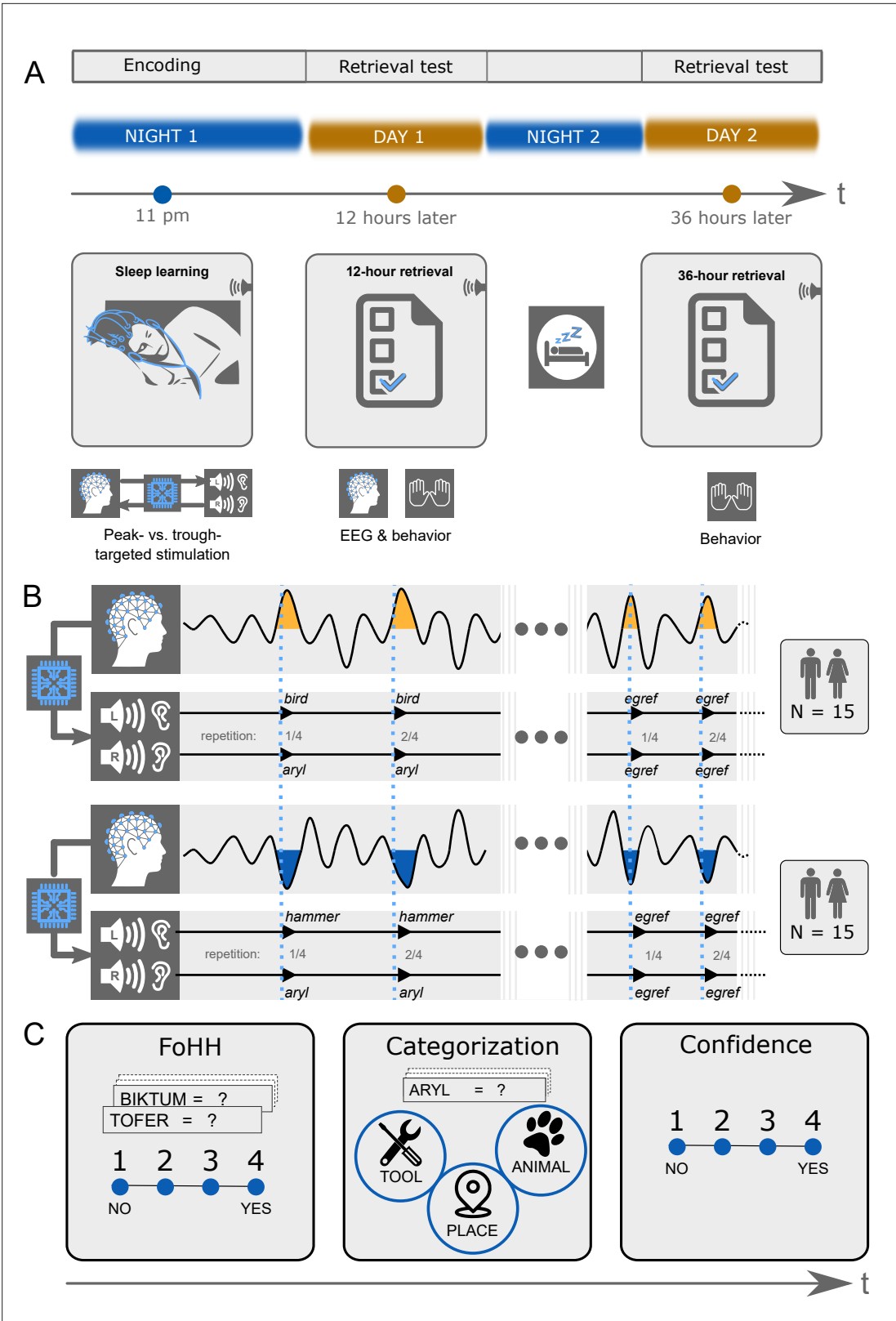

**Figure 1.** Experimental design. (**A**) Design overview: There were three experimental sessions per participant. Thirty participants heard pairs of pseudowords and translation words during sleep starting around 11 pm. They took a first retrieval test at 12 hr and a second retrieval test at 36 hr after the acoustic stimulations during sleep. The EEG was recorded during sleep and during the first retrieval at 12 hr. (**B**) Experimental conditions: We varied between subjects whether the words were played during peaks or troughs of slow-waves. Within subjects we played pairs of pseudowords

*Figure 1 continued on next page*

*Figure 1 continued*

and translation words in the experimental condition and pseudowords alone in the control condition. Both word pairs (experimental condition) and pseudowords (control condition) were presented four times in succession to facilitate sleep-learning. In the experimental condition, translation words were played into the left ear and pseudowords into the right ear. (**C**) Retrieval tasks: Retrieval tasks were the same at the 12 hr and the 36 hr retrieval. Previously sleep-played and new pseudowords were presented at test in both the visual and auditory modality simultaneously (a word appeared on screen and was simultaneously spoken). During each presentation of a pseudoword, participants needed to answer three questions. First, they were asked to indicate whether they had a feeling of having heard (FoHH) the presented word during their sleep in the laboratory. Next, they were asked to assign the presented pseudoword to a superordinate category (animal, tool, place; categorization task). Lastly, they were asked to rate their confidence on a four-point scale regarding their category assignment.

rather than Peaks (main effect Peak versus Trough: $F(1,28) = 5.237$, p=0.030, $d$=0.865, ***Figure 2A***). Retrieval performance did not differ significantly between the two encoding-test delays (main effect Encoding-Test Delay: $F(1,28) = 0.571$, p=0.456). There was no significant interaction between the factor Peak versus Trough and the factor Encoding-Test Delay ($F(1,28) = 0.646$, p=0.428).

Because retrieval performance was significantly better following trough vs. peak targeted stimulation, we computed a second ANOVA for the Trough condition alone (within-subjects factor Encoding-Test Delay). We wanted to determine whether the retrieval accuracy following trough targeting was above chance level (intercept: IV = % correct answers minus mean chance performance) and whether retrieval performance differed between 12 hr and 36 hr (factor Encoding-Test Delay). This ANOVA established a significant intercept: mean retrieval performance was 5.77% above chance level ($M_{Trough}$ = 39.11%, SD = 10.76; $F_{Intercept}$ (1,14)=5.660, p=0.032). Although the 12 hr versus 36 hr comparison was not statistically significant ($F_{Encoding-Test Delay}$ (1,14)=1.308, p=0.272), retrieval performance was numerically larger at 36 hours than at 12 hr ($M_{12hours}$=37.4%, SD = 9.0; $M_{36hours}$=40.7%, SD = 12.4; ***Figure 2BC*** and ***Figure 2—figure supplement 1***). Planned contrasts against chance level revealed that retrieval

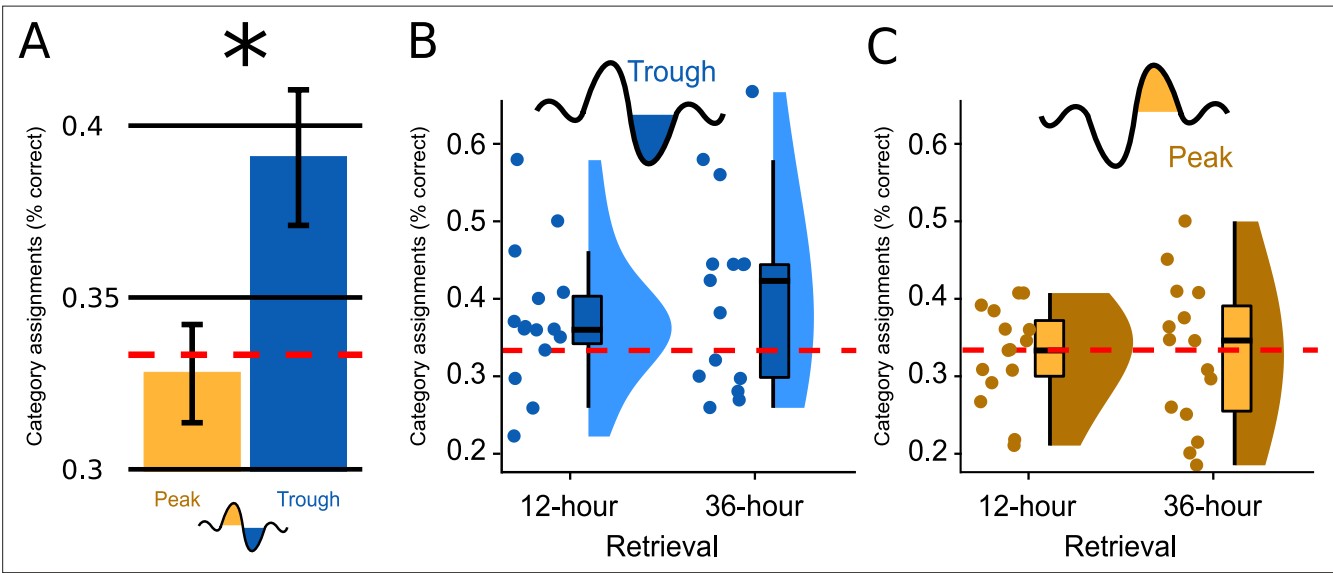

**Figure 2.** Memory performance. (**A**) Bar plots illustrate how well participants had assigned the pseudowords presented at test to a superordinate category in the experimental condition; the data are averaged (*SEM*) over both retrieval time points (N=30). Categorization accuracy in the Trough condition (blue) was above chance (chance = 1/3 correct assignments; $F(1,14)=5.660$, p=0.032) and significantly exceeded the chance-level accuracy of the Peak condition (yellow, $F(1,28) = 5.237$, p=0.030, $d$=0.865). (**B, C**) Boxplot of categorization accuracy split by the Peak/Trough condition and by the 12 hr/36 hr retrieval condition (X-axis). Displayed are box plots with median, 95% confidence intervals, density plots and dot plots of the 15 participant average. The red dotted line indicates chance performance (chance = 1/3 correct assignments).

The online version of this article includes the following figure supplement(s) for figure 2:

**Figure supplement 1.** Paired retrieval performance.

**Figure supplement 2.** Switching rate of category assignment.

**Figure supplement 3.** Feeling of having heard (FoHH) for sleep-paired and new pseudowords.

**Figure supplement 4.** Confidence ratings of category assignments, separated by retrieval accuracy.

performance significantly exceeded chance at 36 hr only ($P_{36hours}$=0.036, $P_{12hours}$=0.094). To better understand why retrieval performance was significantly better in the Trough vs. the Peak condition and why performance exceeded chance level only after 36 hr in the Trough condition, we analysed participants' category assignments at the single-item level. This item-based analysis revealed that participants consistently chose the same category (animals, tools, places) for specific sleep-played pseudowords at the 12- and 36 hr retrieval test (12-to-36 hr consistency rate = 47%, chance level = 33.3%). Moreover, the consistency rate did not differ significantly between the Trough and the Peak condition ($M_{Trough}$ = 47.2%, $M_{Peak}$ = 47.0%, p=0.98). A descriptive analysis of participants' category assignments suggested that the better retrieval performance in the Trough compared to the Peak condition after 36 hr (*Figure 2—figure supplement 2*) was due to the following differences in participants' behaviors:

Participants in the Trough condition were more likely to assign pseudowords to the correct category at 36 hr when they were already correct at 12 hr (Trough: 20% of all items at 36 hr, or 54% of all initially correctly assigned items; Peak: 14% of all items at 36 hr, or 42% of all initially correctly assigned items). (B) Participants in the Trough condition were more likely to switch assignments and use the correct category at 36 hr, if they had initially assigned them to an incorrect category at 12 hr (Trough: 20% of all items at 36 hr, or 32% of the initially incorrectly assigned items; Peak: 18% of all items at 36 hr, or 27% of the initially incorrectly assigned items). Hence, in the Trough condition, pseudowords that were correctly assigned to the object category after 12 hr remained correct after 36 hr, while new correct assignments for initially incorrectly classified pseudowords were added at 36 hr. This is additional evidence that the above-chance retrieval performance at 36 hr is no fluke but originates from sleep-learning.

Because word pairs were exclusively played during slow-wave sleep, which is accompanied by a loss of consciousness (*Dement and Kleitman, 1957*; *Massimini et al., 2012*), encoding and retrieval must have been unconscious. Nevertheless, we assessed whether encoding and retrieval were subjectively unconscious by asking participants to indicate whether they had a "feeling of having heard" (FoHH) the presented pseudoword during sleep (rating from 1: not heard to 4: heard). This wording of the question should set a liberal criterion for reporting having heard a pseudoword. We wanted to obtain information of any potential residual awareness for sleep-played words. Participants' feeling-of-having-heard (FoHH) responses in the Trough condition did not differ significantly between previously sleep-played pseudowords and new pseudowords that had not been played during sleep (see *Figure 2—figure supplement 3*; ANOVA factor 'sleep-played': $F(1,14)$ = 0.45, p=0.51, $M_{sleep-played\_12hours}$=2.41, SD = 0.36, $M_{new\_12hours}$=2.42, SD = 0.353, $M_{sleep-played\_36hours}$=2.53, SD = 0.53, $M_{new\_36hours}$=2.56, SD = 0.618). Furthermore, differences in FoHH responses remained non-significant even if we compared only the correctly categorized pseudowords to the never heard, new pseudowords ($F(1,14)$ = 0.08, p=0.78). We also found no significant difference between the FoHH for correct versus incorrect category assignments (ANOVA factor 'accuracy': $F(1,14)$ = 0.09, p=0.77, $M_{12hours\_correct}$=2.37, SD = 0.54, $M_{12hours\_incorrect}$=2.42, SD = 0.39, $M_{36hours\_correct}$=2.57, SD = 0.62, $M_{36hours\_incorrect}$=2.49, SD = 0.55) and no difference between EC and CC trials (ANOVA factor "Association": $F(1,14)$ = 0.09, p=0.77, $M_{12hours\_correct}$=2.37, SD = 0.54, $M_{12hours\_incorrect}$=2.42, SD = 0.39, $M_{36hours\_correct}$=2.57, SD = 0.62, $M_{36hours\_incorrect}$=2.49, SD = 0.55). The same was true for the Peak condition, where we did not find significant differences in the FoHH reports (*Figure 2—figure supplement 2*; sleep-played vs new: $F(1,14)$ = 1.027, p=0.33; correct vs incorrect EC trials: $F(1,14)$ = 0.04, p=0.85, EC vs CC: $F(1,14)$ = 0.04, p=0.85). These findings strongly suggest that participants could not consciously recognize sleep-played pseudowords.

Furthermore, we explored whether participants' confidence ratings revealed an unconscious reminiscence of sleep-played word pairs for correctly versus incorrectly categorized items at retrieval. Hence, we included the confidence scale as a subtle measure of memory besides the cruder accuracy measure. At retrieval testing, participants rated their confidence for each assignment of a pseudoword (sleep-played and new) to a superordinate word category (animal, tool, place). Confidence ratings did not differ signifcantly for sleep-played (EC and CC) compared to new pseudowords in the Trough condition, were sleep-learning was successful ANOVA factor 'sleep-played': $F(1,14)$ = 0.47, p=0.5; $M_{sleep-played\_12hours}$=2.17, SD = 0.38, $M_{new\_12hours}$=2.19, SD = 0.376, $M_{sleep-played\_36hours}$=2.26, SD = 0.426, $M_{new\_36hours}$=2.26, SD = 0.426. Furthermore, confidence ratings for correct versus incorrect category assignments in the experimental condition differed neither in the Trough condition (*Figure 2—figure supplement 4*), ANOVA factor 'Accuracy': $F(1,14)$ = 2.36, p=0.15; $M_{12hours\_correct}$=2.24, SD = 0.42,

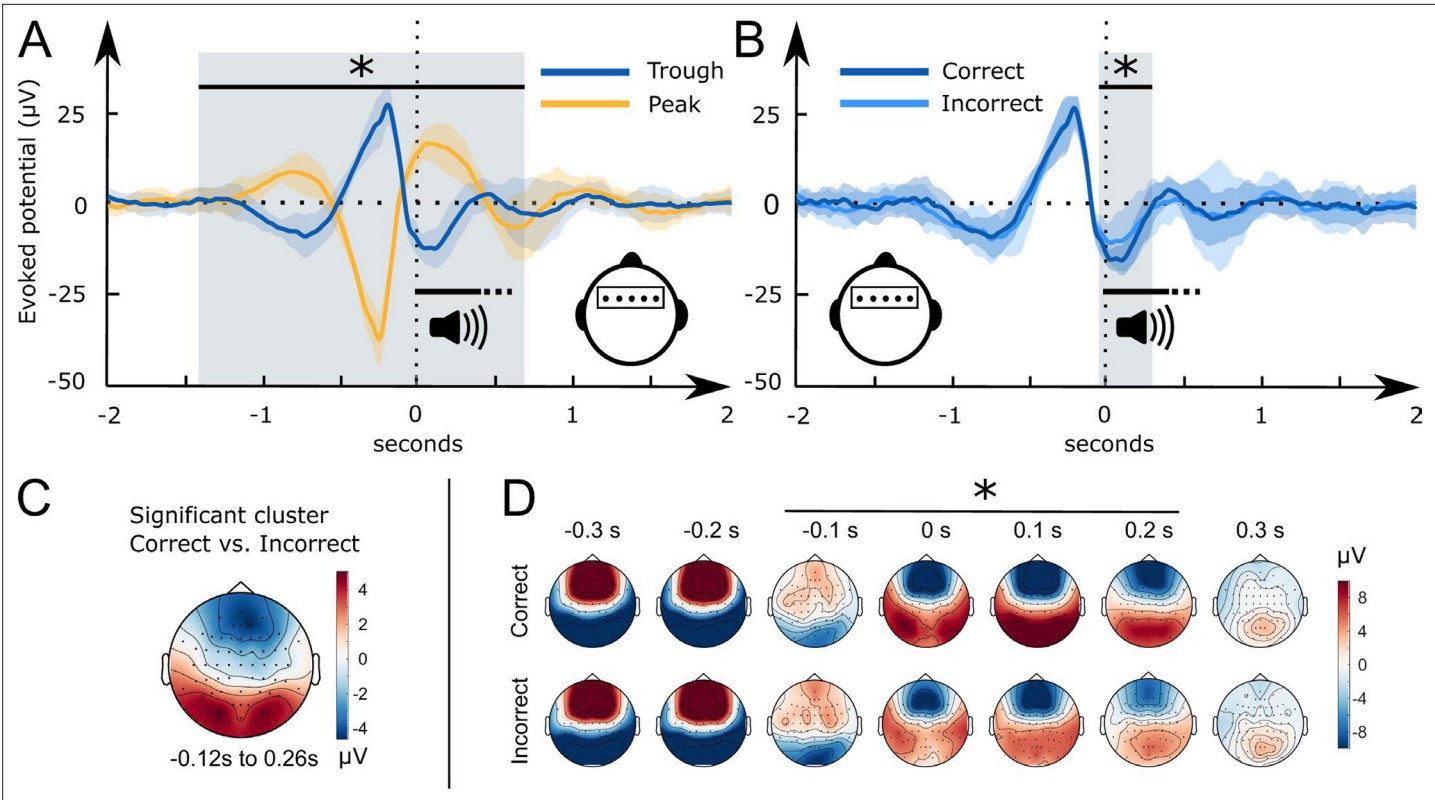

**Figure 3.** Word-related EEG potentials recorded during sleep. (**A**) Word pair-related voltage response plotted for the Trough condition (blue) and the Peak condition (yellow, *SD* is shaded). (**B**) Comparison of subsequently correctly (dark blue) and incorrectly (light blue) assigned pseudowords to categories at the 36 hr retrieval in the Trough condition (N=15). (**C**) Topographical voltage distributions of the averaged significant cluster for the contrast of correct versus incorrect category assignments at the 36 hr retrieval. (**D**) Time course of the topographical voltage distribution for correctly and incorrectly assigned pseudowords at the 36 hr retrieval. * indicates significant time points; cluster-level Monte Carlo p<0.05. Shaded areas represent the standard deviation between subjects. All trials are sorted with reference to the word onset (0 seconds, vertical dotted line).

The online version of this article includes the following figure supplement(s) for figure 3:

**Figure supplement 1.** Accuracy-related evoked EEG potentials recorded during sleep.

$M_{12hours\_incorrect}$=2.16, SD = 0.35, $M_{36hours\_correct}$=2.33, SD = 0.44, $M_{36hours\_incorrect}$=2.28, SD = 0.4 nor the Peak condition (*Figure 2—figure supplement 4*), ANOVA factor Accuracy: $F(1,14)$ = 0.48, p=0.50; $M_{12hours\_correct}$=2.19, SD = 0.51, $M_{12hours\_incorrect}$=2.28, SD = 0.54, $M_{36hours\_correct}$=2.33, SD = 0.58, $M_{36hours\_incorrect}$=2.36, SD = 0.51.

## Word-evoked potentials differed initially between the peak and trough condition and then aligned between conditions from 700 ms to 2 s following word onset

The EEG data were recorded in the sleeping participants to determine differences in the processing of the played words between conditions. We analysed the electrophysiological responses to word pairs in the Trough versus the Peak condition locking the electrophysiological response to the onset of word pairs. The scalp topographies of the event-related potentials (ERPs) recorded in Troughs versus Peaks were almost diametrically opposed frontal cluster [–1.42 s to 0.48]: cluster-level Monte Carlo *P*<0.002; two occipital clusters [–1.38 s to–0.09s] and [0.07s to 0.71s]: cluster-level Monte Carlo for both clusters p<0.002; *Figure 3A*. This difference attests to the successful targeting of peaks versus troughs by the used brain-state-dependent stimulation algorithm. For further manipulation checks of the stimulation algorithm see methods. The word-evoked response in the Peak condition resembled the response to white noise clicks in slow-wave entrainment studies (*Andrillon and Kouider, 2020*; *Cox et al., 2014b*; *Ngo et al., 2013*). An early frontal positivity at 200 ms was followed by a large negative component at 550 ms, which in turn was followed by a late positivity at 900 ms (*Figure 3*). These components

have been described as the generic response of the sleeping brain to any kind of sensory stimulation (*Andrillon and Kouider, 2020*; *Cox et al., 2014b*; *Laurino et al., 2014*; *Laurino et al., 2019*; *Riedner et al., 2011*). *Halász, 2016* suggested that these components resemble K-complexes that reflect the brain's response to maintaining sleep in the presence of sensory stimuli.

However, when we targeted words to troughs, the played words evoked two positive frontal components. The first positive component appeared at 500 ms and the second at one second following word onset. Hence, the second component corresponded to the entrained late positivity in the Peak condition. In short, targeting words to peaks evoked a generic response that presumably blocked the encoding of sleep-played word pairs in the Peak condition. This is underlined by recent findings of *Niknazar et al., 2023*, who demonstrated that the large negative component of the evoked K-complex inhibits long-range communication in the brain. Targeting words to troughs shifted the EEG and inhibited this large negative component. The ERP differences between the Trough and the Peak condition vanished at 700 ms. Between 700 ms and 2 s following word onset the EEG was comparable between the two conditions.

## A pronounced frontal trough promoted word processing

We computed word-evoked potential differences measured during sleep for those items that were correctly versus incorrectly assigned to the three superordinate categories at 36 hr. We focused on behavior at the 36 hr retrieval interval because performance was nominally higher at this vs. at the 12 hr interval. Results for the 12 hr interval were qualitatively similar (for details see supplement).

Starting 50 ms before and ending 260 ms following word onset, frontal and occipital electrodes recorded an amplified voltage (larger frontal negativity and larger occipital positivity) for subsequently correctly versus incorrectly assigned pseudowords in the Trough condition (*Figure 3B*, frontal cluster: cluster-level Monte Carlo p=0.016 at −50 ms to 260 ms; occipital cluster: cluster-level Monte Carlo p=0.010 at −50 ms to 260 ms). In the Peak condition, there was no significant difference between subsequently correctly versus incorrectly assigned pseudowords (*Figure 3—figure supplement 1* cluster-level Monte Carlo p>0.53). Furthermore, there was no significant difference between the EC and CC trials in the Peak (cluster-level Monte Carlo p=0.54) and the Trough condition (cluster-level Monte Carlo p>0.37). The difference map between topographies for subsequently correctly versus incorrectly assigned pseudowords in the Trough condition resembled the voltage distribution of a frontal slow-wave trough (see *Figure 3C*, voltage distribution of a prototypical trough). Hence, sleep-learning in the Trough condition benefited from a pronounced trough during stimulus onset.

## A brain-wide voltage distribution that is typically associated with a frontal trough promoted word processing

The visual inspection of the difference in the topographical voltage distribution at word onset between subsequently (36 hr) correctly versus incorrectly assigned pseudowords (*Figure 3C*) suggested that sleep-learning in the Trough condition was best, if the brain-wide voltage distribution corresponded to the typical trough state as quantified with a template of a slow-wave trough. Note that we targeted slow-wave troughs/peaks by correlating the online measured EEG with a template map (*Ruch et al., 2022*). The template map is based on pre-recorded EEG data (*Züst et al., 2019*) and represents the average voltage distribution of thousands of peaks/troughs (peak template, trough template). Accordingly, we consider the trough template a prototypical voltage distribution of endogenously generated troughs. Descriptively, the measured voltage map underlying later (at 36 hr) correctly versus incorrectly assigned pseudowords corresponded to the template map of a typical trough (*Figure 3C*). Therefore, we tried to find out, which trough aspect had promoted sleep-learning. We examined four trough aspects measured at word onset: (A) prototypicality reflecting the correspondence (i.e. correlation) of the trough voltage distribution at word onset with the prototypical voltage distribution of a trough, that is the trough template. A high prototypicality suggests that the trough originated in the same neocortical network that generates the majority of endogenous slow waves (*Michel and Koenig, 2018*) (B) global field power (GFP) reflecting the synchronisation within the measured trough voltage map. A high GFP indicates strong coherence and stability of neuronal activity within the cortical network that generates the trough (*Khanna et al., 2015*) (C) inter-trial phase coherence (ITC) reflecting the temporal coherence of Trough-targeting across the four presentations of a specific word pair; (D) time difference reflecting the time difference between the actual acoustic stimulation

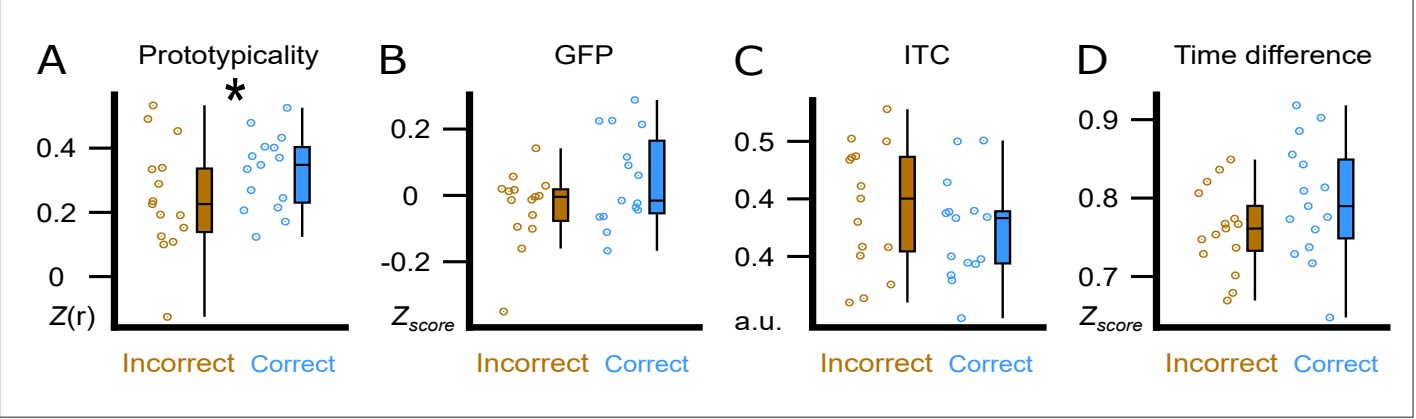

**Figure 4.** Trough features compared between correct and incorrect category assignments at the 36 hr retrieval. We display the comparison between correctly (blue) and incorrectly (brown) assigned pseudowords in the experimental condition at the 36 hr retrieval. (**A**) Using the amplitude of the correlation between the ERP and the template (Prototypicality, Fisher's z) at word onset. (**B**) Using global field power (GFP, z-score) of the ERP at word onset. (**C**) Using the inter-trial phase coherence (ITC, a.u.) between the four presentations of a word pair. (**D**) Using the time difference (Time Difference, z-score) between the actual acoustic stimulation and the measured trough maximum. Participant averages (dots), group median (line) and 95% confidence intervals (whiskers) are displayed as boxplots. * indicates a significant difference, p=0.005 (paired t-test, N=15).

The online version of this article includes the following figure supplement(s) for figure 4:

**Figure supplement 1.** Trough features compared between correct and incorrect category assignments at the 12 hr retrieval.

and the measured trough maximum (*Figure 4*; for a more detailed description see method section). Prototypicality differentiated significantly between correctly (higher prototypicality) versus incorrectly assigned pseudowords at 36 hr (p=0.005, FDR corrected: $q_{\text{Benjamini-Hochberg}}$ = 0.0125; *Figure 4A*). This comparison remained significant when we compared the area under the prototypicality curve (AUC; p<0.019,–150 ms to 350 ms, FDR corrected: $q_{\text{Benjamini-Hochberg}}$ = 0. 02). AUC was computed to ensure that not only the word onset, but the entire trough period was more similar to the trough template for subsequently correctly versus incorrectly assigned pseudowords. The other trough aspects did not differentiate significantly between subsequently correctly versus incorrectly assigned pseudowords (all p>0.13). The same Trough-related analyses for correctly versus incorrectly assigned pseudowords at 12 hr are presented in the supplement (*Figure 4—figure supplement 1*). These analyses were not performed for the Peak condition because retrieval performance was not better than chance in the Peak condition.

## Enhanced theta and fast spindle power promoted sleep-learning

We aimed to isolate neural markers of semantic-associative encoding from word onset to 2.5 s thereafter. To this aim, we compared EEG responses to the presentation of word pairs (EC) to responses to presentations of pseudowords (CC) during sleep. Pseudowords were presented alone (no added translation word) into both ears in the control condition and were played in alternating order with word pairs (EC, CC, EC, CC, EC,...). Trough-targeted word pairs versus pseudowords enhanced the theta power at 500 ms following word onset (0.2 s to 0.7 s, cluster-level Monte Carlo p=0.018, no significant cluster for Peak-targeted word pairs, p>0.1, *Figure 5A*). Trough-targeted word pairs versus pseudowords enhanced the fast spindle power at 1 s following word onset (0.8 s-1.2 s, cluster-level Monte Carlo p=0.012, no significant cluster for Peak-targeted word pairs, p>0.4, *Figure 5B*). The average theta enhancement in the significant cluster correlated significantly with retrieval performance (theta: *R*=0.57, p=0.027, *Figure 5A*), while the spindle enhancement did not correlate with retrieval performance (spindle: *R*=0.034, p=0.9, *Figure 5B*). The comparison between correctly and incorrectly assigned pseudowords did not yield any significant cluster in the theta- or spindle-band power following word onset (window: 0–2.5 s, cluster-level Monte Carlo $p_{Theta}$ >0.6; $p_{Spindle}$ >0.1).

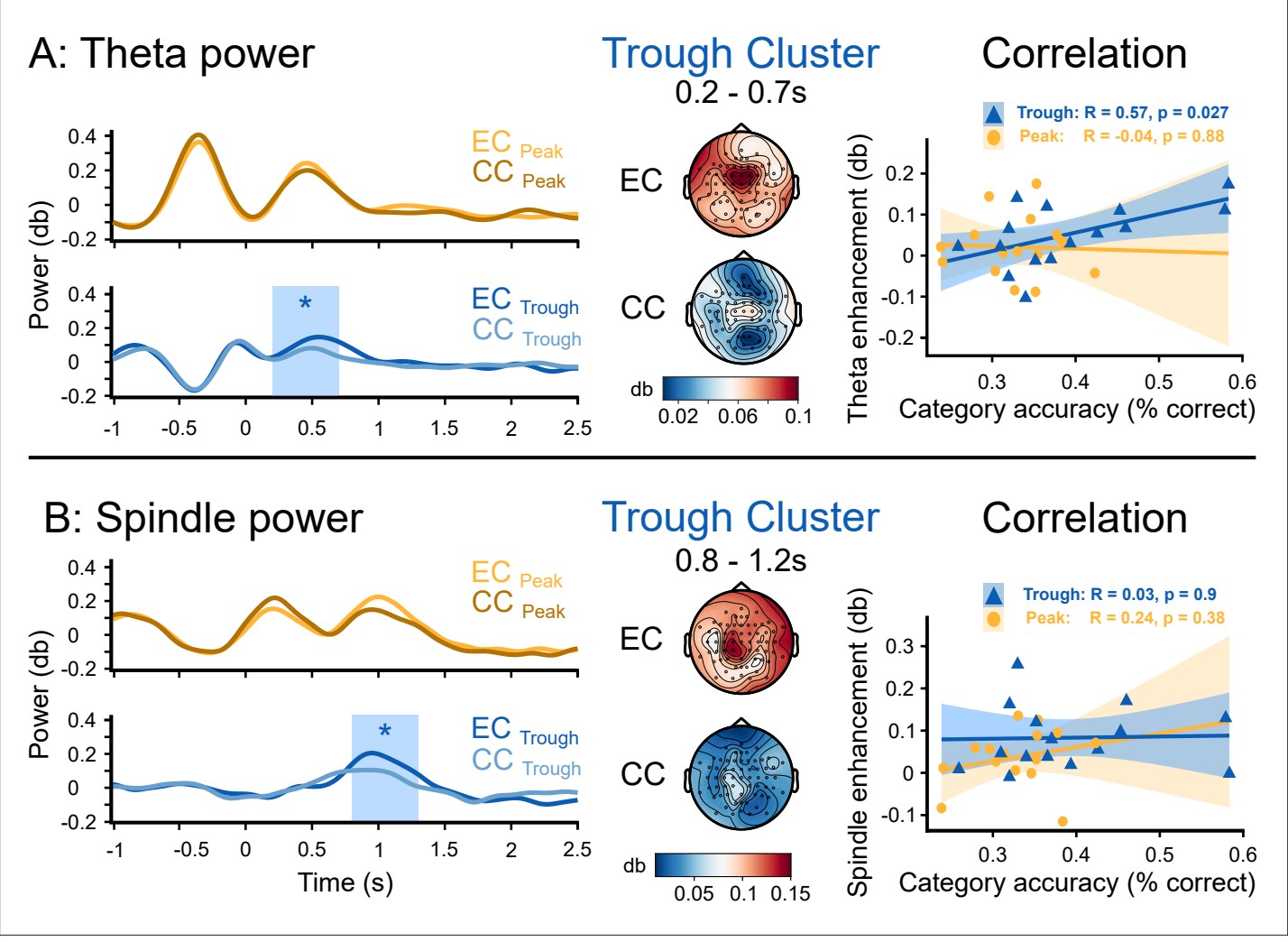

**Figure 5.** Theta and fast spindle power following word onset. Time course of the topographical distribution of theta (**A**) and fast spindle (**B**) power, shown for the experimental condition (EC), where word pairs were played, and for the control condition (CC), where pseudowords were played (left, average of representative electrodes). The blue box demarcates clusters that reflect a significant difference between EC and CC (*: cluster-level Monte Carlo p<0.05, N=15). Topographical distribution of the theta and the fast spindle power averaged over the significant time period and electrode displayed for the experimental condition (EC, top) and the control condition (CC, bottom) in the Trough condition. Correlation of the theta enhancement and the fast spindle enhancement measured from the cluster of significant electrodes with the individual accuracy of category assignments in the experimental condition at the 36 hr retrieval (right).

### Increased neural complexity in the trough condition mirrors word processing

To further identify potential neural markers of semantic-associative encoding during sleep, we examined how Peak- vs. Trough-targeted stimulation altered the post-stimulus neural complexity of the EEG signal. Measures of neural complexity provide non-linear estimates of the variability and 'randomness' of the EEG signal and indicate the amount of information available in the signal at each channel. Neural complexity is indicative of both the brain's capacity to process sensory information (*Waschke et al., 2017*; *Waschke et al., 2019*) and the extent of ongoing cognitive activity (*Höhn et al., 2023*; *Parbat and Chakraborty, 2021*). Furthermore, levels of neural complexity closely correspond to the depth of sleep (*Casali et al., 2013*; *Höhn et al., 2023*; *Ma et al., 2018*; *Türker et al., 2023*), with the lowest levels typically observed during SWS. Within sleep, periods of elevated neural complexity are associated with a heightened propensity to process verbal information. *Andrillon and Kouider, 2016* reported larger lateralized readiness potentials indicating correct semantic categorization of words during sleep when words were played during states of elevated neural complexity. *Türker*

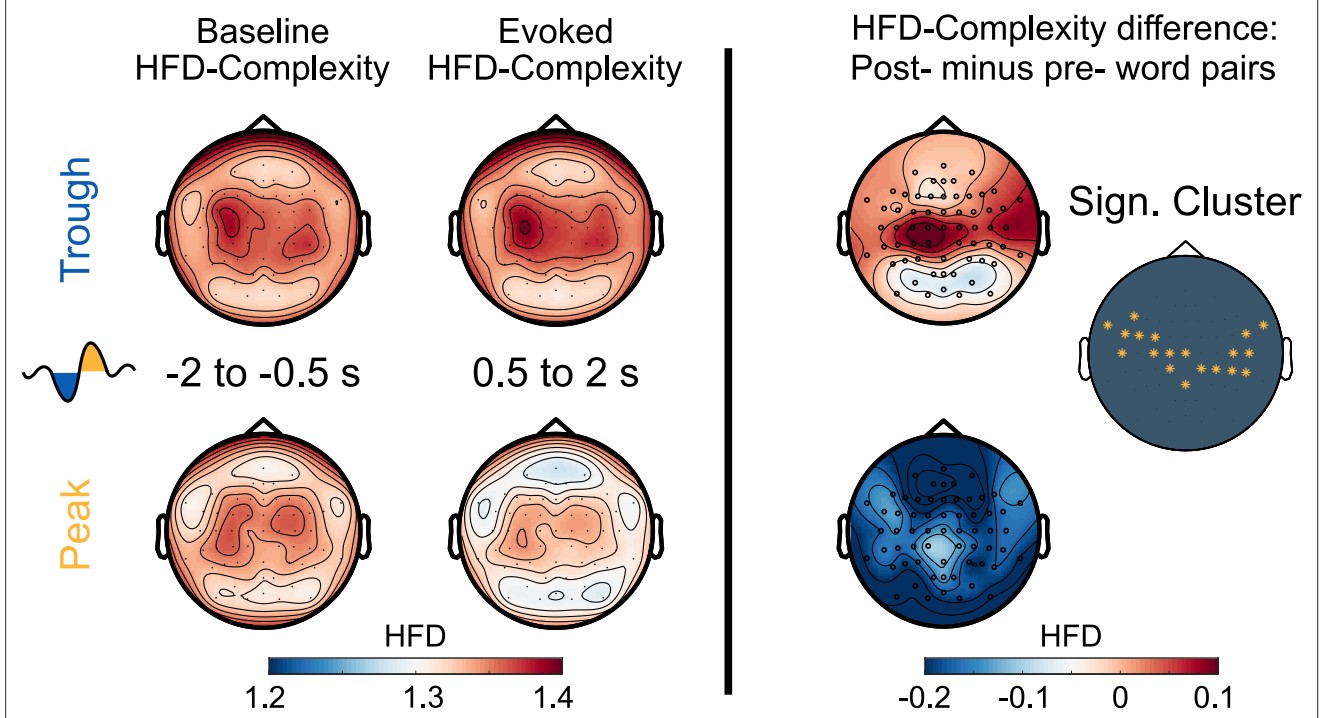

**Figure 6.** Neural complexity. EEG-derived neural complexity measured as Higuchi Fractal Dimension (HFD) displayed for the Trough condition and the Peak condition before the presentation of word pairs (Baseline) and following the presentation of word pairs (Evoked) in the experimental condition. The right panel displays the significant difference between the Trough and the Peak condition in the neural complexity gain from before to following word onset (*cluster-level Monte Carlo: p=0.002, N=15).

*et al., 2023* further observed that sleeping participants were more likely to produce accurate facial responses to words vs. pseudowords when stimulus presentations occurred during states of high complexity. Importantly, auditory stimulation during sleep tends to temporarily reduce post-stimulus neural complexity (*Alnes et al., 2024*; *Andrillon et al., 2016*), probably reflecting an inhibitory, sleep-protective brain response to stimulation.

Here, we explored whether word presentations would have distinct impacts on post-stimulus neural complexity when stimulation was targeted to the Trough, where sleep-learning occurred, vs. to the Peak, where no sleep-learning occurred. To this aim, we calculated the Higuchi Fractal Dimension (HFD) of the EEG signal in the time domain at each channel separately for a pre- (−2 to –0.5 s) and a post-stimulus (0.5–2 s) time window. HFD is one of many measures of neural complexity (*Lau et al., 2022*). We analyzed HFD for a late post-stimulus window ranging from 0.5 s up to 2 s because we assumed that neural complexity for this time-window would not be biased by the differences that are due to the targeted state (Peak/Trough) but would mirror stimulus-induced neural activity. Note that for this time-window, the word-evoked EEG potentials no longer showed differences between Peak versus the Trough condition.

Baseline corrected EEG complexity following word offset (from 500 ms to 2 s, when the EEG was aligned between the two conditions) was higher in the Trough than the Peak condition (cluster-level Monte Carlo p=0.002; *Figure 6*). However, we observed that presentation of word pairs reduced post-stimulus complexity compared to the pre-stimulus baseline in the Peak but not the Trough condition (post-stimulus complexity compared to baseline; Peak condition: cluster-level Monte Carlo p=0.002; Trough condition: cluster-level Monte Carlo p>0.1). The Peak-specific reduction in complexity might reflect the brains' standard inhibitory response to sensory stimulation during sleep (*Andrillon et al., 2016*; *Höhn et al., 2023*). Importantly, this inhibitory response was absent following Trough-targeted stimulation. In fact, post-stimulus complexity was descriptively (see *Figure 6*) but not significantly increased compared to baseline in the Trough condition. This absence of a post-stimulus dip in neural complexity could reflect learning-associated neural activity in the Trough condition (*Lau et al., 2022*; *Parbat and Chakraborty, 2021*). However, baseline-corrected HFD did not differ significantly

between word pairs that were later (at 12 hr and at 36 hr) correctly versus incorrectly assigned to categories (correct vs incorrect Peak and Trough: all cluster-level Monte Carlo p>0.49). Nor did baseline-corrected HFD significantly differ between the EC and the CC (EC vs CC Peak and Trough: all cluster-level Monte Carlo p>0.16). Finally, we correlated HFD values with the participants' accuracy of assigning pseudowords to the three categories at the 36 hr retrieval of the Trough condition. This correlation yielded an insignificant result (*R*=0.25, p=0.36). In conclusion, word processing during sleep appears to increase neural complexity in troughs.

## Discussion

Natural slow-wave sleep is a state in which our conscious awareness of the surrounding world is drastically reduced. For our protection, we still need to process external events in order to decide whether an event is threatening and requires an immediate awakening or whether an event is relevant enough to warrant storage for later consideration. There is evidence indicating that we can process and store new information during slow-wave sleep (*Ai et al., 2018*; *Andrillon and Kouider, 2016*; *Arzi et al., 2012*; *Arzi et al., 2014*; *Koroma et al., 2022*; *Ruch et al., 2014*; *Ruch and Henke, 2020*; *Züst et al., 2019*). However, whether the most sophisticated form of learning - episodic learning - is possible during deep sleep, is contentious. This experiment reveals that troughs and peaks of slow-waves contribute mutually to successful episodic learning during sleep. Vocabulary played during troughs of slow-waves passed the thalamic gate and was processed perceptually and conceptually, which raised neural complexity. The following peak harbouring an increase in theta power provided the conditions for semantic-associative encoding of pseudowords and translation words. An immediately following second peak accompanied by a rise in fast spindle power around 1 s following word onset may have aided the immediate consolidation of the formed associations. This sequence of processing steps prepared the ground for later stages of memory consolidation that resulted in a successful retrieval of associations after 36 hr.

Was the sleep-presented vocabulary stored and retrieved through the episodic memory system? Traditionally, episodic memory and hippocampal processing were considered processes of declarative or explicit memory and were thought to require conscious awareness of the learning material (*Gabrieli, 1998*; *Moscovitch, 2008*; *Schacter, 1998*; *Squire and Dede, 2015*; *Tulving, 2002*). Yet, increasing counterevidence indicates that hippocampal-assisted episodic memory formation may proceed with and without conscious awareness of the learning material (*Duss et al., 2014*; *Henke et al., 2003*; *Henke et al., 2013*; *Wuethrich et al., 2018*; *Züst et al., 2015*). The conscious and unconscious rapid encoding and flexible retrieval of novel relational memories was found to recruit the same or similar neural networks including the hippocampus (*Henke et al., 2003*; *Schneider et al., 2021*). This suggests that conscious and unconscious episodic memories are processed by the same memory system. In the current study, we directed word processing to the episodic memory system by: (a) exacting *associative encoding*, (b) enforcing a *speedy encoding process*, and (c) compelling a delayed cued *associative retrieval* that requires a flexible representation of the sleep-formed memories. These task-enforced processing characteristics are key features of episodic memory with no other memory system disposing of these computational abilities (*Cohen and Eichenbaum, 1993*; *Henke, 2010*; *O'Reilly et al., 2014*). Only the hippocampus can form arbitrary word-word associations and rapidly store the associations in a flexible format. The neocortex forms new arbitrary associations only slowly over dozens of learning trials and provides for fused (rather than flexible) word-word representations in memory (*Cohen and Eichenbaum, 1993*; *Henke, 2010*; *O'Reilly et al., 2014*). Although no brain imaging was performed in the current study, evidence that relational learning during sleep is mediated by the hippocampus was provided by findings of *Züst et al., 2019*. Using a very similar memory task, these authors observed that successful retrieval of sleep-played associations was associated with increased hippocampal activity. Mere priming is unlikely to account for the observed learning effects because priming is not compositional and flexible enough to provide the cued retrieval of word meaning that is required by the retrieval task. Note that at retrieval, only the pseudoword was presented. Hence, participants had to infer the corresponding category by first retrieving the sleep-played translation word and then classifying the translation word's semantics to the superordinate category (tool, animal, place). We reason that memory representations formed by priming are not compositional and flexible enough to promote the required cued retrieval of word meaning. In summary, the task requirements called upon the episodic memory system and previous evidence

(*Züst et al., 2019*) of hippocampal activation during correctly versus incorrectly categorized items at wake-retrieval testing and the between-subjects correlation of this contrast with retrieval performance suggest that the sleep-formed memories were mediated by the episodic memory system.

While consciously acquired information in the episodic memory system tends to decay rapidly during the 24 hr following learning (*Ebbinghaus, 2013*; *Hardt et al., 2013*; *Murre and Dros, 2015*), we did not observe a deterioration in retrieval performance from 12 hr to 36 hr. This speaks to the longevity and robustness of sleep-formed memories. Because participants enjoyed a night of undisturbed sleep at home between the 12 hr and the 36 hr retrieval, the newly formed associations may have undergone additional sleep-assisted memory consolidation. One might argue that the retrieval task after 12 hr led to a conscious re-encoding of sleep-formed memories. However, the sleep-played translation words were not presented at test, and participants received no feedback regarding the accuracy of their responses during the retrieval task. Therefore, it seems unlikely that the successful delayed retrieval after 36 hr was due to conscious re-learning during the first retrieval. Instead, the reactivation of sleep-formed memories during the first retrieval task might have tagged these memories for further (re-) consolidation during the ensuing night (*Rabinovich Orlandi et al., 2020*).

In fact, sleep-dependent memory consolidation supports weak memories more than strong memories (*Denis et al., 2020*; *Drosopoulos et al., 2007*; *Petzka et al., 2021*; *Schechtman et al., 2021*; *Schneider et al., 2021*; *Tucker and Fishbein, 2008*). Hence, the presumably weak sleep-formed memories in this study may have profited especially from sleep-dependent memory consolidation. We have recently found that unconscious memories formed from subliminal (consciously invisible) cartoon clips (instead of memory formation during sleep) benefited from sleep-dependent memory consolidation, such that they were retrievable after hours (*Pacozzi et al., 2022*).

However, before memory consolidation can kick in, the sleep-played vocabulary needs to pass thalamic gating and undergo neocortical processing (*Gent et al., 2018*; *McCormick and Bal, 1994*). How was this possible? We played the vocabulary either during peaks or during troughs of slow-waves. The retrieval performance indicated that the vocabulary was processed only in the Trough condition. When troughs were pronounced, encoding and retrieval were best, and the EEG exhibited more neural complexity reflecting cognitive processing (*Parbat and Chakraborty, 2021*). We offer two speculative explanations as to why the processing of the played vocabulary was possible during troughs rather than peaks of slow-waves. First, the peaks may have been 'occupied' by the consolidation of previously wake-formed memories. Wake-formed memories appear to be preferentially reactivated during peaks, when hippocampus-neocortical interactions take place, which are accompanied by a rise in fast spindle power (*Göldi et al., 2019*; *Mölle et al., 2002*; *Mölle et al., 2011*; *Muehlroth et al., 2019*; *Staresina et al., 2015*). Even a causal role of endogenous spindles is suggested for memory consolidation (*Antony et al., 2019*; *Chen and Wilson, 2017*; *Latchoumane et al., 2017*; *Maingret et al., 2016*; *Mölle et al., 2011*; *Staresina et al., 2015*), particularly if the spindles are nested into peaks (*Dang-Vu et al., 2011*; *Schabus et al., 2012*). Troughs, on the other hand, may not be occupied by ongoing consolidation processes and might therefore be receptive to sounds. Second, targeting vocabulary to troughs bypasses the natural sleep preserving mechanisms at play during peaks of slow-waves. Any sound evokes a systematic endogenous EEG response, often described as a K-complex (*Andrillon and Kouider, 2020*; *Cox et al., 2014b*; *Halász, 2005*; *Halász, 2016*; *Mölle et al., 2009*; *Ngo et al., 2013*; *Schabus et al., 2012*). Such a generic response represents a cortical sensory gate that protects sleep and the consolidation of wake-formed memories (*Andrillon and Kouider, 2020*; *Halász, 2005*). The prominent frontal negativity at 500 ms in particular reflects a breakdown of the cortico-thalamic communication (*Niknazar et al., 2023*). This breakdown of communication appears to leave sensory processing intact in primary sensory cortices but mitigates responses at higher cortical levels (*Schabus et al., 2012*). When words were played into peaks, we recorded a K-complex like, generic response from frontal electrodes, while when words were played into troughs, the vocabulary evoked a phase reset of the slow-waves that differed sharply from K-complexes. To conclude, we assume that ongoing consolidation processes and sleep-protective responses generated during peaks of slow-waves inhibited the psycholinguistic processing of the simultaneously played pseudowords and translation words, while these words were psycholinguistically processed during troughs.

Because the cued associative retrieval was above chance following sleep-learning in the Trough condition, the translation words must have been understood and their meaning must have been

bound to the sound of the pseudoword during slow-wave sleep. We assume that the psycholinguistic processing of the translation word lasted up to 500 ms, which was also the duration of the word utterance and the duration of the ongoing trough. The brain starts extracting the meaning of a word already while it is being uttered. The phonological, syntactic, lexical, and semantic word analyses occur in parallel during the 500 ms after word onset (*Brodbeck and Simon, 2022*; *Hagoort, 2008*; *Hickok and Poeppel, 2007*; *Pulvermüller et al., 2009*; *Skeide and Friederici, 2016*). ERP and MEG recordings during (awake) spoken word processing revealed that phonological word forms are processed 20–50 ms following word onset in the auditory cortex. Phonological word forms are categorized at the morphosyntactic level at 40–90 ms. Lexical–semantic word analyses occur at 50–80 ms in the left anterior superior temporal cortex, where lexical items associated with the phonological word forms are retrieved at 110–170 ms. These same temporal areas funnel lexical information to Brodmann areas 45 and 47, where semantic relations between words are analysed between 200–400 ms (*Skeide and Friederici, 2016*). The timing of external stimulus processing during slow-wave sleep appears to be similar to the waking state (*Laurino et al., 2019*; *Ruby et al., 2008*; *Sabri et al., 2000*). Awake learning experiments in humans indicated that the hippocampus ramps up its encoding machinery at 300–500 ms after word onset (*Long et al., 2014*; *Quiroga et al., 2005*; *Staresina and Wimber, 2019*). The perceived associations between the meaning of translation words and the sound of pseudowords need to be encoded in the hippocampus (*Sakaguchi and Hayashi, 2012*; *Tonegawa et al., 2018*) that induces long-term potentiation and increases the numbers of dendritic spines to strengthen the connectivity between neurons (*Bliss and Collingridge, 1993*; *Engert and Bonhoeffer, 1999*; *Hebb, 1949*). This process of memory formation requires the collaboration of the hippocampus with neocortical networks (*Buzsáki and Tingley, 2018*; *Schreiner and Staudigl, 2020*). Hippocampal associative binding depends on theta activity (and theta-gamma coupling) in the hippocampus (*Axmacher et al., 2006*; *Fernández-Ruiz et al., 2017*; *Kahana et al., 2001*; *Mormann et al., 2005*; *Osipova et al., 2006*). In the current study, theta power rose at 500 ms following word onset. This might reflect the moment when semantic pseudoword-word associations underwent storage in the hippocampus. The magnitude of the increase in theta power correlated, between-subjects, with retrieval performance at 36 hr. One interpretation is that theta activity supported the association of pseudowords with translation words. Importantly, at 500 ms following word onset, a slow-wave peak followed the stimulated trough, which provided for the necessary neuronal excitability and an effective hippocampal-neocortical communication (*Andrillon and Kouider, 2020*; *Destexhe et al., 2007*). The frontally recorded peaks coincided with enhanced theta power, and both may have contributed to a hippocampal encoding of new associations. Initially labile hippocampal-neocortical memory traces are usually replayed to secure storage by consolidation (*Goto et al., 2021*; *Rasch and Born, 2013*). Fast sleep spindles support this process (*Chen and Wilson, 2017*; *Latchoumane et al., 2017*; *Maingret et al., 2016*; *Petzka et al., 2022*; *Schreiner and Staudigl, 2020*; *Staresina et al., 2015*) because they constitute a 'shuttle' of hippocampal representations to neocortical sites (*Antony et al., 2019*; *Cairney et al., 2018*). We assume that fast sleep spindles supported the immediate replay of the sleep-formed associations at one second following word onset (similar to *Abdellahi et al., 2023*). This was the time when a second peak occurred and when fast spindle power ramped up in the experimental condition but not in the control condition, where no associative encoding took place because pseudowords alone were played during troughs. Both the increase in theta power at 500 ms and the increase in fast spindle power at one second were accompanied by an enhancement of neuronal activity. These neuronal events were absent if the vocabulary was played during peaks. Hence, critical events that foster learning and storage were observed in the Trough condition, where vocabulary was stored long-term, but not in the Peak condition, where no sleep-learning was observed.

According to the sequence of events described above, we speculate that learning during sleep may proceed as follows: (1) A frontal slow-wave trough allows for spoken language comprehension by providing a 500 ms time window for external stimulus processing (no sleep-protective K-complexes). (2) 500 ms following word onset, paired-associative encoding takes place during the next slow-wave peak, coincident with enhanced theta power, both providing for the necessary hippocampal-neocortical crosstalk. (3) 1000 ms following word onset, fast spindles assist the immediate replay of the encoded information through hippocampal-neocortical interactions. This model needs to be tested in future sleep-learning experiments using magnetoencephalography or the simultaneous recording of functional magnetic resonance imaging data and EEG data for a precise temporal

mapping of the processing stages and the simultaneous measurement of associated activations in brain regions.

While the current experiment is the first to target acoustic stimuli to slow-wave peaks and troughs for episodic learning during sleep, we have earlier performed a similar experiment playing vocabulary at fixed intervals (open-loop) during slow-wave sleep (*Züst et al., 2019*). In that study, the pseudoword and translation word of a pair were played in succession rather than simultaneously. This changes the timing and the sequence of processing steps. With words played in succession (*Züst et al., 2019*), the paired-associative encoding in memory cannot happen before the second word of a pair is being played. In the current study, the two words of a pair were simultaneously presented and the paired-associative encoding in memory occurred around 500 ms following the onset of both words. Because hippocampal paired-associative encoding requires broadly activated neurons for an effective hippocampal-neocortical connectivity (*Cox et al., 2014b*; *Destexhe et al., 2007*; *Schabus et al., 2012*; *Sirota and Buzsáki, 2005*), a slow-wave peak at the time of associative learning seems mandatory. This time point is during the play of the second word of a pair, when word presentations are sequential, and the time point is around 500 ms following word onset, when word presentations are simultaneous. The critical paired-associate encoding process required a frontally recorded slow-wave peak in both the *Züst et al., 2019* study and the current study. It should be noted that the comparison of the *Züst et al., 2019* study with the current study is difficult because *Züst et al., 2019* entrained the sleep-EEG by rhythmically playing words inter-stimulus-interval 1053 ms; *Ngo et al., 2013*, which changed the natural course of the sleep EEG. Therefore, only the current study permits conclusions regarding the role of endogenously generated slow-wave peaks and troughs in mediating sleep-learning.

Because slow-wave sleep is a state of unconsciousness, a sophisticated form of learning like rapid vocabulary acquisition is not expected given the still prevailing dogma of episodic learning depending on conscious awareness (*Gabrieli, 1998*; *Moscovitch, 2008*; *Schacter, 1998*; *Squire and Dede, 2015*; *Tulving, 2002*). Rapid vocabulary acquisition during deep sleep adds to evidence of successful episodic learning from subliminal (consciously invisible) words presented in the waking state (*Duss et al., 2014*; *Henke et al., 2013*; *Reber et al., 2012*). These findings support the claim that episodic memory formation may proceed without conscious awareness (*Dew and Cabeza, 2011*; *Hannula and Greene, 2012*; *Henke, 2010*). The sleep-formed memories did not significantly subside but (non-significantly) strengthened over 36 hr to the point where they influenced deliberative decision-making. Evidence of language processing and rapid verbal associative learning during deep sleep challenges the views that slow-wave sleep is a state of general synaptic depotentiation (*Rasch and Born, 2013*; *Tononi and Cirelli, 2006*), that we have a strict gating of sensory information at the thalamus, and that the direction of information flow during sleep is strictly from hippocampus to neocortex (*Eban-Rothschild et al., 2016*; *Rasch and Born, 2013*).

Finally, we would like to acknowledge certain limitations inherent to our research design, methodology, and interpretation. First, our interpretation of how learning may have occurred during sleep is speculative and hypothetic at this point. The delineated sequence of word processing steps during slow-wave phases should inspire future research aimed at testing these hypotheses. Second, sound duration was a bit longer than the average duration of the targeted slow-wave phases (500 ms) and therefore extended into the following slow-wave phase. This lowers the specificity of the contrast between the slow-wave phases. Third, there is a discrepancy in auditory complexity and information density between the experimental and control conditions (pseudoword-word pairs vs. one single pseudoword). These differences can induce processing and EEG differences between conditions other than associative learning. Fourth, carry-over effects from the first to the second retrieval test are possible and may have produced the slight increase in retrieval accuracy from the first to the second retrieval.

Future studies might look into practical applications as well as the ethical problems (*Stickgold et al., 2021*) of acoustic stimulations during sleep. While the acquisition of new vocabulary is certainly best during waking, other forms of sleep-learning might yield better results, when learning is unconscious versus conscious. Sleep-learning might be exploited by entities with nefarious intentions (*Stickgold et al., 2021*). However, sleep-learning might also be put to good use, such as in psychotherapy. Because sleep-learned messages are processed and stored unconsciously, they circumvent conscious defence mechanisms (*Arzi et al., 2014*; *Levy et al., 2014*). This information might dispose the sleeper

more readily towards behavioural change than traditional psychotherapy conducted in the waking state. Finally, sleep-learning may allow for the reactivation and the subsequent modification and reframing of unwanted memories without the need to consciously re-experience the stressful memories (*He et al., 2015*; *Taschereau-Dumouchel et al., 2018a*; *Taschereau-Dumouchel et al., 2018b*; *Zhu et al., 2022*).

# Methods

## Contact for resource sharing

Further information and requests for resources should be directed to the corresponding author Flavio Schmidig, E-mail: flavio.schmidig@gmail.com.

## Experimental model and subject details

### Participants

We aimed for a sample size of N=30 participants based on a previous sleep-learning study (*Züst et al., 2019*). To achieve this sample size, we had to examine a total of 68 participants. Of these participants, 34 could not sleep long enough for the experimenter to play the vocabulary during slow-wave sleep. In these participants, we discontinued the experiment before the behavioural tests the next morning. In four additional participants, our brain-state-dependent stimulation algorithm failed. These four participants were thus also excluded from analysis. The final analyses included the data of 30 participants (age: 19–28, *M± SD* = 24.16±2.32; 22 [73.3 %] female), whereof 15 were assigned to the Trough condition and 15 to the Peak condition. All included participants were right-handed and reported normal hearing abilities and the absence of mental and physical illness. They denied a history of sleep disorders and declared pursuing a regular sleep schedule. The participants' sleep during the night preceding the night at the sleep laboratory was restricted to five hours. Adherence to this sleep restriction was assessed via self-report. Moreover, participants refrained from naps and consuming stimulants (e.g. coffee) the day before the night at the sleep laboratory.

Participants were fully informed of the study protocol and of the fact that the study includes vocabulary learning during sleep. Participants also knew that the retrieval of the sleep-learned vocabulary would be tested during the next two mornings following the stimulation night. However, participants were naïve regarding their assignment to the Trough or Peak condition, and regarding the simultaneous, lateralized presentation of pseudowords (right ear) and translation words (left ear). We informed participants about sleep-learning because we wanted to provide a convincing explanation for why participants had to wear in-ear headphones during the entire night and why not removing them was essential. We hoped that the information about sleep-learning would unconsciously prime participants for processing the sounds during sleep without waking up. Prior to experimentation, participants gave their written consent to the study protocol, which had been approved by the local ethics committee (Kantonale Ethikkommission Bern, 2017–01046).

## Method details

### Experimental design

The experimental design included one between-subjects factor 'Peak versus Trough': the vocabulary was either played during peaks (15 participants) or during troughs (15 participants) of slow-waves. The experimental design also included two within-subjects factors, namely the factor 'Encoding-Test Delay' with two levels (12 hr, 36 hr) and the factor 'Stimulation' with two levels (experimental condition, control condition). In the experimental condition, we played vocabulary binaurally. Vocabulary consisted of pseudowords that were played to the right ear, and German translation words that were played to the left ear. In the control condition, we played pseudowords binaurally (one pseudoword was simultaneously played to both ears).

### Stimuli

A total of 96 pseudowords and 36 German translation words were used in this study. Nine of the 36 German translation words were presented in practice trials that participants took before experimentation to become familiar with the forced-choice semantic categorization task used for retrieval testing. The pseudowords were adopted from *Züst et al., 2019*. They consisted of two-syllabic pseudowords

(no meaning) and were originally created by combining German and Dutch syllables (*Duyck et al., 2004*; *Züst et al., 2019*). The spoken pseudowords lasted around 600 ms (mean = 0.591 s, SEM = 0.006 s). The German translation words were also two-syllabic and exhibited approximately the same speech duration (mean = 0.541 s, SEM = 0.011 s). Each German translation word was prototypical for one of three superordinate categories: animals, tools, places. These categories were chosen because of their neuroanatomically distinct brain activation patterns. Distinct brain activation patterns are useful for category decoding based on the recorded EEG. The German translation words exhibited similar lexical frequencies (according to Leipzig Corpora Collection) throughout categories.

Because the retrieval test required participants to assign each pseudoword to one of three super-ordinate categories (animals, tools, places), we made sure that the sound of the spoken pseudowords was not indicative of a certain category; that is we avoided sound symbolism to influence category assignments (*Sidhu and Pexman, 2017*). To this aim, we asked 80 students to indicate for each spoken pseudoword whether it represents an animal, a tool, or a place. We then selected those 96 pseudowords from a large set that showed the least category bias. Furthermore, we computed post-hoc analyses on the data acquired in the retrieval tests of the Trough condition to find out whether any remaining sound symbolism had systematically biased participants' category assignments. The number of correct assignments at the 36 hr retrieval in the Trough condition (where sleep-learning was demonstrated) was not significantly different between the three categories ($F(2,28)$ = 0.77, p=0.47). We also computed an ANOVA with the two independent within-subject variables 'Category' (animals, tools, places) and 'Correctness of Choice' (correct, incorrect) and with the dependent variable 'Number of Choices' for the 36 hr retrieval in the Trough condition. This ANOVA yielded neither a significant main effect of Category ($F(2,28)$ = 0.39, p=0.67) nor a significant interaction between Category and Correctness of Choice ($F(2,28)$ = 0.49, p=0.615). Only the main effect Correctness of Choice reached significance ($F(1,14)$ = 8.40, p=0.012) reflecting participants' above chance performance. In sum, sound symbolism exerted no significant influence on category assignments.

Audio-files of all stimuli were processed manually to have equal loudness. This was done to eliminate potential loudness artefacts on the EEG-signal. Salient phonetic features (plosive and sibilant sounds) were attenuated manually. All audio-files were compressed for dynamic range and normalised for peak volume (as in *Züst et al., 2019*).

A set of 27 German translation words and 27 pseudowords was used for sleep-learning in all participants. German translation words and pseudowords were randomly combined to pairs for each participant. Hence, any pseudoword could be combined with any German translation word. This procedure reduced the risk that potentially remaining category-biases exerted by pseudowords would systematically influence retrieval accuracy. An additional set of 27 pseudowords was used in the control condition, where a pseudoword (without a translation word) was played to both ears. Hence, words presented in the control condition contained no meaning. The 27 pseudowords presented in the experimental condition and the 27 pseudowords presented in the control condition were later represented at the 12 hr and the 36 hr retrieval for their assignment to categories. In addition, we presented 21 pseudowords that had not been played during sleep at the 12 hr retrieval. At the 36 hr retrieval, these same 21 pseudowords were represented as well as an additional set of 21 entirely new pseudowords. Participants assigned these non-sleep-played pseudowords also to categories. Yet, these assignments were considered neither correct nor incorrect because no translation words had been associated with these pseudowords.

## Procedure

Participants arrived at the sleep laboratory at 10 p.m. and were equipped with EEG electrodes. They filled in several questionnaires, namely the Pittsburgh Sleep Quality Index (*Buysse et al., 1989*), the Stanford Sleepiness Scale (*Hoddes et al., 1973*), and a sleep diary. Next, we determined the participant's hearing threshold to adjust the sound volume individually for the subsequent presentation of vocabulary during sleep. An hour after arrival, participants went to bed wearing in-ear headphones for vocabulary presentation during sleep. When lights went out, we administered a relaxation exercise, progressive muscle relaxation (*Jacobsen, 1929*; *Kalra et al., 2015*) to facilitate sleep onset.

The experimenter waited for the participant to fall asleep. Once the recorded EEG showed stable slow-wave sleep, the experimenter started the auditory stimulation. Word presentation was controlled by a brain-state-dependent stimulation algorithm, which detected upcoming slow-wave peaks and

troughs (*Ruch et al., 2022*). Words were either played during peaks or troughs of slow-waves, depending on the experimental condition the participant was randomly assigned and blind to. The experimenter monitored the participant's sleep EEG and paused the auditory stimulation if the EEG displayed signs of arousal. Auditory stimulation was resumed once a participant was back in stable slow-wave sleep. Following the play of all words (27 * 4 word pairs in the experimental condition and 27 * 4 pseudowords in the control condition; total of 216 presentations), participants were woken up and were sent home to continue their night sleep.

In the next morning, participants visited the sleep laboratory again between 11 a.m. and 2 p.m., that is exactly 12 hr following their recorded slow-wave sleep phase. Participants were again outfitted with EEG electrodes and took the 12 hr retrieval task, while their EEG was being recorded. Nine practice trials familiarized participants with the retrieval procedure. Twenty-four hours later, participants returned again to the laboratory to perform the 36 hr retrieval, which was administered without EEG registration.

Following the 36 hr retrieval, we applied a formal hearing test to ensure that each participant's hearing ability was in the normal range regarding volume and frequencies. For this hearing test, we played short beeps at random intervals targeting the left or right ear. The participants' task was to indicate, which ear was stimulated. We varied the frequency of the played tones, the tone's volume, and the type of tone (pink noise and pure tones; frequency: 500 Hz, 1000 Hz or 2000 Hz; 15–45 dB). All participants exhibited hearing abilities in the normal range.

## Auditory stimulation during sleep

When the experimenter observed stable slow-wave sleep for at least 2 min (4 sleep scoring time windows) in the EEG, he started auditory stimulation. To habituate participants to the presence of auditory stimuli, we first played white-noise bursts (instead of words). The volume of these white-noise burst was at first below the individual hearing threshold and was then raised to the participant-specific, predetermined volume (~35 dB). The target volume was adjusted to the hearing threshold for each participant. A sound-proof chamber with very low background noise (~30 dB) and in-ear headphones allowed to play the sounds at such a low volume. Once the target volume was reached with the participant remaining sound asleep, the experimenter initiated the presentation of the words that belonged to the experimental (e) and the control (c) condition. The sequence of the sleep-played words was systematically alternated between conditions (E-C-E-C-E-C...; *Figure 1B*). Words within condition were presented in an order that was randomly generated for each participant. Each word pair (experimental condition) and pseudoword (control condition) was presented four times in direct succession before the next word pair/pseudoword was presented. The experimenter monitored the participant's sleep and paused auditory stimulation whenever the EEG indicated arousal. The experimenter resumed the auditory stimulation once the EEG return to stable slow-wave sleep.

Participants were played a mean of 101.23 (SD = 2.40) word pairs (maximum: 4x27 = 108) in the experimental condition. The number of word pairs played depended on the duration of a participant's slow-wave sleep phase. We post-hoc excluded word pairs from the analysis, if one or more of the four presentations of a word pair had occurred outside slow-wave sleep or if a presentation was accompanied by a muscle artefact within a time window of +/-3 s of stimulation onset. Therefore, a final mean of 95.33 (SD = 15.41) word pairs entered data analysis. This final number was not significantly different between the Peak and the Trough condition ($M_{Trough}$ = 95.2 ($SD_{Trough}$ 17.11), $M_{Peak}$ = 96.8 ($SD_{Peak}$ 13.43); $t$ (26.5)=–0.285, p=0.778).

## Retrieval task

To probe participants' memory of the sleep-played word pairs, we gave them three tasks that they completed in immediate succession on each presented pseudoword. The pseudowords were presented simultaneously visually (on a monitor) and acoustically (as during sleep). The presentation order of the pseudowords was randomly generated for each participant (i.e. it varied between participants). When participants logged their response, the program progressed to the next task.

Upon the presentation of a pseudoword, participants were first prompted to indicate any 'feeling of having heard' (FoHH) the presented pseudoword during sleep. They responded with a rating between 1 and 4 (1=no feeling of having heard, 4=strong feeling of having heard). We asked participants to indicate whether they feel that they may have heard the word during sleep. We wanted participants

to set a liberal criterion for reporting a feeling of having heard a word because we were interested in any semi-conscious or conscious word recognition. Then, we prompted participants to guess whether the presented pseudoword designates an animal, a tool, or a place. This task was intended to reactivate a sleep-formed pseudoword-translation word association and to trigger the conversion of the reactivated translation word (e.g. dog) into the superordinate semantic category (e.g. animal). Mean chance performance on this task was 33.33% correct responses. Participants chose one of the three categories by pressing a keyboard button. The assignment of keyboard buttons to categories (animal, tool, place) was randomly shuffled between the response trials to exclude habituation and hence a category-associated motor response pattern in the neuronal data. Finally, we prompted participants to indicate on a four-point-scale how confident they were about their previous category assignment. This metacognitive evaluation was to reveal any conscious/semi-conscious hunch of a previously sleep-formed associative memory.

We probed the participants' memory twice, first at 12 hr and then again at 36 hr following the vocabulary presentation during sleep. At the 12 hr retrieval, we presented 75 pseudowords, whereof 27 had been sleep-played along with a translation word in the experimental condition, 27 were sleep-played (without translation words) in the control condition, and 21 were presented for the first time. All of these pseudowords were presented to each participant, although not all pseudowords entered the data analysis for each participant because some pseudowords could not be played during sleep to certain participants. The 36 hr retrieval proceeded exactly like the 12 hr retrieval. Yet, it included 21 additional pseudowords that had not been presented at the 12 hr retrieval. Therefore, we presented 96 pseudowords at the 36 hr retrieval.

## Equipment
### Sleep laboratory
We used two computers in this experiment. One computer was used for the recording of the EEG and the other for the control of the auditory stimulation. The EEG data were streamed in real-time via LAN from the recording computer to the stimulation computer. The stimulation computer hosted the brain-state-dependent stimulation algorithm that controlled auditory stimulation during sleep. The stimulation computer was also used to administer the retrieval tasks and to conduct the hearing test. We programmed the retrieval task using the software Presentation (Neurobehavioral Systems (http://www.neurobs.com), version 23). The brain-state-dependent stimulation algorithm was implemented in MATLAB 2017. We used the Psychophysics Toolbox of MATLAB (**Brainard, 1997**) for the sound presentation. Commercial in-ear headphones (Pioneer, type SE-CL502_L) delivered the auditory stimulation to the participant. For each presentation, a TTL trigger was sent from the stimulation computer to the recording computer to mark stimulation onset times in the EEG for later analyses. Triggers were sent over the digital input output board "U3" by LabJack U3 (https://labjack.com/products/u3).

### Polysomnography and sleep scoring
We recorded 64-channel EEG with a customised 10–20 montage using BrainCap MR BP-03010MR with 'Fast'n Easy' electrodes (http://www.easycap.de) and two BrainAmp DC, MR plus 32 channel amplifiers by Brain Products (http://www.brainproducts.com). We recorded the EEG with BrainVision Recorder (http://www.brainproducts.com). Ground was set at CPZ electrode, reference at Fz electrode. Two additional electrodes were placed laterally beneath the eyes to record eye movements (EOG). One electrode at the chin recorded the muscle tone (EMG). The EEG sampling rate was 500 Hz. Impedances were kept below 20 kΩ.

Sleep scoring was performed according to the guidelines of the American Academy of Sleep Medicine (AASM; **Iber et al., 2007**). For the online detection of the onset of slow-wave sleep and for the surveillance of slow-wave sleep, the real-time EEG was displayed according to the AASM guidelines using the OpenViBE environment (**Renard et al., 2010**). Offline sleep scoring was performed by two trained raters, who were blinded to the experimental conditions, with the software Polyman (http://www.edfplus.info/). The interrater reliability reached a Cohen's Kappa of 0.83, which is substantial (**McHugh, 2012**). In cases, where the two raters disagreed, we used the more conservative rating, i.e., the sleep score that indicated the shallower sleep stage. The processing of the EEG data was performed with the MATLAB toolboxes EEGLAB (http://sccn.ucsd.edu/eeglab/, version, version 14.1.2b) and fieldtrip (https://www.fieldtriptoolbox.org/, version 20190819). If the offline scoring

revealed that a word pair or word was played outside slow-wave sleep, all repetitions of this stimulus were excluded from data analysis for this subject.

The 30 participants slept for a mean of 59.3 min ($SEM$ 24.11 min) before the experimenter woke them. They spent a mean of 32.3 min in slow-wave sleep ($SEM$ 12.70 min). Total sleep time did not differ between the Peak-targeted and the Trough-targeted participant sample (Peak: $M_{Peak}$ = 67.4 min, SEM = 27.2 min; Trough: $M_{Trough}$ = 51.1 min, SEM = 18.0 min, $t$=–1.929, p>0.065). The duration of slow-wave sleep was significantly shorter for the Trough-targeted participant sample than for the Peak-targeted participant sample (Trough: $M_{SWS}$ = 26.3 min, $SEM_{SWS}$ = 15.7, Peak: $M_{SWS}$ = 38.2 min, $SEM_{SWS}$ = 14.0; $t$=–2.9, p=0.01). Furthermore, the inter-trial stimulus interval (ISI) happened to be shorter in the Trough than the Peak condition (Peak: $M_{ISI}$ = 6.3 s, $SD_{ISI}$ = 2.6 s; Trough: $M_{ISI}$ = 3.3 s, $SD_{ISI}$ = 1.1 s; p<0.003). These differences are due to the fact that the brain-state-dependent stimulation algorithm targeted troughs better than peaks. Peak-to-trough transitions are more prominent EEG features than trough-to-peak transitions because peak-to-trough transitions have higher amplitudes and steeper slopes. The brain-state-dependent stimulation was more likely to detect these peak-to-trough transitions and therefore targeted more troughs than peaks within the same amount of time. Consequently, the ISI was shorter in the Trough than the Peak condition. For this reason, we were able to present the vocabulary within a narrower SWS time-span and could wake and release the participants earlier in the Trough than the Peak condition. Importantly, neither the length of the ISI nor the duration spent in slow-wave sleep (both z-transformed; N=30) correlated with retrieval accuracy at 36 hr (ISI: $r$=–0.017, p=0.93; slow-wave sleep duration: $r$=–0.11, p=0.56). Therefore, the length of the ISI and the duration spent in slow-wave sleep do probably not account for the difference in retrieval performance between the Trough and the Peak condition. Of note, the number of words played during slow-wave sleep did not differ between the Peak and the Trough condition (Peak: $M_{Peak}$ = 24.2, SD = 3.6; Trough: $M_{Trough}$ = 24.1, SD = 4.3; p>0.95).

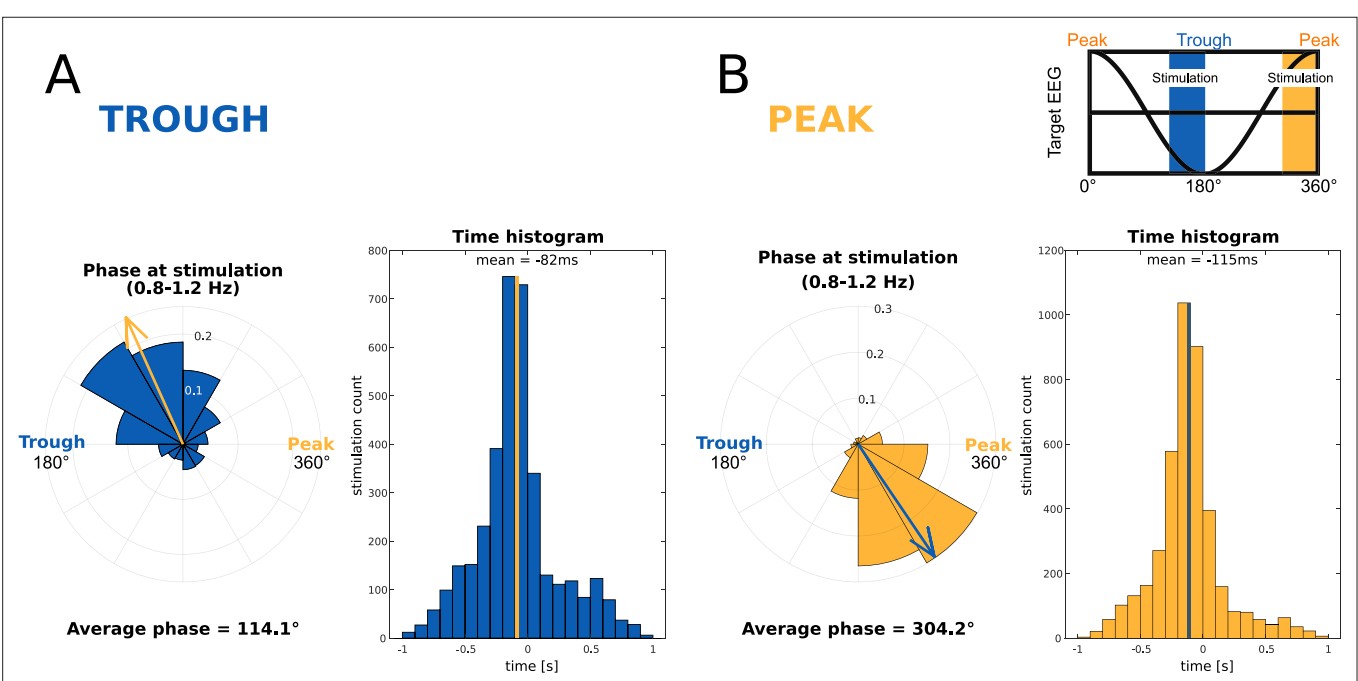

**Figure 7.** Accuracy of the targeting of slow-wave phase. Utterance into a half-wave of a slow-wave. The mean word onset was 113 ms before peak maxima (all stimulations). Data showing the precision of phase targeting for slow-wave Troughs (**A**, N=15, blue) and Peaks (**B**, N=15, yellow). On the left side in A and B, we present rose plots that indicate the phase of slow-wave filtered voltage (08.–1.2 Hz). The arrows demarcate the average phase of acoustic stimulation; 360° represents a peak and 180° represents for a trough. On the right side in A and B, we present the histogram illustrating the time delay between a real peak/trough and the acoustic stimulation. The highlighted line indicates the average of all trials. On the top right is a translation of slow-wave phases into angles, colors depict the targeted slow-wave phases. We targeted the acoustic stimulation ahead of the local maxima (peak or trough) to fit the entire word utterance into a peak/trough. The onset of word presentation was on average 82 ms or 66° before the trough maximum and 115 ms or 54° before the peak maximum.

## Brain-state-dependent targeting of slow-wave peaks and troughs

Peaks and troughs of slow-waves were targeted based on the correlation of the online registered scalp EEG map with a template of a prototypical slow-wave (*Ruch et al., 2022*). We targeted the onset of the spoken words to a time point of 100 ms before the peak/trough maxima/minima in order to fit each word utterance into a half-wave of a slow-wave. The mean word onset was 113 ms before peak maxima and 96 ms before trough minima (*Figure 7*). Thus, the onset of spoken words preceded a trough on average by 64.05 degrees 95% CI = [53.15 74.95] and a peak by 56.8 degrees 95% CI = [54.08 59.52], *Figure 7*. The targeting accuracy of the used the brain-state-dependent stimulation algorithm was comparable to other algorithms in the field (*Cox et al., 2014b*; *Ngo et al., 2013*). The phase distribution in both the Peak and the Trough condition was significantly non-uniform $p_{Peak}$ <0.001, $p_{Trough}$ <0.001; Hodges-Ajne omnibus-tests (*Zar, 1999*) as implemented in the CircStat toolbox (*Berens, 2009*). This indicates that our brain-state-dependent stimulation algorithm targeted peaks and troughs reliably. This stimulation accuracy allows attributing differences in sleep-learning/wake-retrieval between the Peak and the Trough condition to these same brain states rather than to technical fuzziness.

## Quantification and statistical analysis

### Data analysis

Behavioural data were analysed in R (https://www.r-project.org/, version 2022.07.0) using the packages dplyr, tidyr, ggplot2, ez, wesanderson, lme4, reshape2, readxl, rstatix, and RColorBrewer. For the analysis of the EEG data, we used the MATLAB toolboxes EEGLAB (http://sccn.ucsd.edu/eeglab/, version, version 14.1.2b) and fieldtrip (https://www.fieldtriptoolbox.org/, version 20190819) as well as custom made scripts. The error probability of 5% was chosen for all statistical tests. Where appropriate, we corrected the p-values for false discovery rate in multiple hypothesis testing using the Benjamini-Hochberg method.

### Behavioural data analysis

The outcome measure for sleep-learning was wake-retrieval accuracy at 12 hr and at 36 hr. Participants assigned the previously sleep-played and new pseudowords to one of three categories (animal, tool, place). If the assigned category corresponded to the superordinate category of the translation word that had been played during sleep along with the pseudoword, then the participant's response was classified as correct. Participants' assignments of new pseudowords to the three categories were neither correct nor incorrect because new pseudowords were not associated with a certain translation word. The new pseudowords were included at the 12 hr and the 36 hr retrieval as a baseline for the event-related EEG potential analyses (correct versus incorrect versus baseline).

Categorization accuracy for previously sleep-played pseudowords was expressed as percentage of correct responses. If accuracy was significantly above the mean chance level of 33.33%, we inferred successful sleep-learning. Accuracy values of participants did not significantly deviate from a normal distribution at the 12 hr and at the 36 hr retrieval ($p_{Shapiro}$ >0.148), variances did not significantly violate the assumption of homogeneity ($p_{Levene}$ >0.34), and all values were within the range of +/-2 standard deviations from the mean. We computed a 2x2 ANOVA with the between-subjects factor Peak versus Trough and the within-subjects factor Encoding-Test Delay (12 hr versus 36 hr). The dependent variable was retrieval accuracy expressed as the difference between the percentages of correctly retrieved associations minus the percentage of mean chance performance (33.33 %). A second ANOVA with the within-subjects factor Encoding-Test Delay (12 hr versus 36 hr) was computed for the Trough condition alone to determine whether the intercept is significant (indicating that the average retrieval accuracy exceeded chance level in the two encoding-test delays) and to determine whether retrieval performance was different at 12 hr versus 36 hr. Furthermore, we planned a comparison of retrieval performance against chance level at the 12 hr and at the 36 hr retrieval.

## EEG data analysis

### EEG preprocessing

The raw EEG-data were re-referenced to the common average, band-pass filtered at 0.25–35 Hz, and down-sampled to 100 Hz. The continuous EEG data was subjected to manual artefact rejection. First,

noisy channels and channels with strong sweat artefacts were visually identified and subsequently interpolated. Next, we visually identified and excluded data segments that contained muscle artefacts, movement artefacts, or arousals. Artefact rejection was performed by Flavio Schmidig, who was blinded to the participants' condition (Peak/Trough) and to the type and timing of sound presentation.

## Event-related EEG analyses

We computed sleep-recorded event-related potentials (ERPs) and event-related spectral perturbations (ERSPs) for each participant. For ERPs, we first epoched the raw EEG data into 7 s trials (from 3 s before to 4 s after word onset) and then averaged over all trials per condition and per participant. Moreover, we compared ERSPs between experimental conditions. To this aim, we computed the spectral power for the fast spindles (12–16 Hz) and the theta band (4–8 Hz). We estimated the power over the interval of 2 s before to 3 s after word onset using discrete wavelet transformation (20 ms steps, i.e. 50 Hz sampling rate). We used linearly increasing cycle numbers with 4.25 cycles at the lowest and 11.75 cycles at the highest frequency for the theta frequency band and with 4.25 cycles at the lowest and 11.75 cycles at the highest frequency for the spindle frequency band. ERSPs were baseline-corrected at the single trial level by normalising the time-series at each frequency using the frequency-specific mean and standard deviation of the power, which was computed across the entire trial (*Grandchamp and Delorme, 2011*). Baseline-corrected ERSPs were then averaged over all trials of a condition. This procedure was performed for each electrode, each condition, and each participant. Then, we computed the grand averages for the conditions (Peak condition, Trough condition, experimental condition, control condition, correct assignments in the experimental condition, incorrect assignments in the experimental condition). To test for statistical differences, we performed mass univariate analyses with cluster-based permutation statistics (1000 permutations) to correct for multiple comparisons (*Maris and Oostenveld, 2007*). Cluster-level Monte Carlo values were considered significant if error probability was lower than 5%.

## Extraction of slow-wave characteristics

Our aim was to find out which slow-wave characteristics promoted sleep-learning. Therefore, we computed the standardized global field power of the targeted slow-wave (GFP, z-score) at the time of word onset in each trial. We averaged over the four presentations of a word pair and over each participant. Furthermore, we computed the Peak/Trough prototypicality, which reflected the similarity (Fisher's z-transformed correlation values) between the electrical field (scalp maps) of the online-recorded EEG at word onset and the electrical field of the targeted slow-wave phase (Peak/Trough). Next, we extracted the inter-trial-phase coherence at word onset to estimate the consistency of the targeted slow-wave phase across the four presentations of each word pair (measured over frontal electrodes, a.u.). Finally, we computed the z-scored time delay between word onset and the time-point of the maximal amplitude of the targeted slow-wave peak/trough. We compared these parameters between correctly versus incorrectly assigned pseudowords at the 12 hr and the 36 hr retrieval. Where necessary, the data were z-transformed using Fisher's transformation.

## Offline detection of slow-wave peaks and troughs

In order to identify the delay between the time of word onset and the time of the maximal amplitude of the targeted slow-wave peak/trough, we post-hoc identified discrete slow-wave events in the EEG. Slow-waves were automatically detected by a MATLAB-based algorithm modelled after *Mölle et al., 2002*. Slow-wave peaks and troughs produce distinct brain wide voltage distributions (*Ruch et al., 2014*), which we used to target online slow-wave phases. For the off-line identification of slow-wave peaks and troughs, we used frontal derivatives. We bandpass-filtered the EEG from 0.5 to 1.5 Hz and averaged the EEG over the frontal electrodes ('F1', 'F2', 'F3', 'F4', 'Fz'). Next, we identified every zero-crossing in the bandpass filtered frontal signal. The local minima and maxima between zero-crossings were labelled as potential peaks and troughs. In this process, time and amplitude constraints were added. All slow-waves with durations shorter than 0.8 s or longer than 2 s were excluded. The resulting slow-waves lay between 0.5 and 1.25 Hz. A further criterion was slow-wave amplitude. The detected slow-wave half-waves needed to exhibit an amplitude (trough-to- peak) that exceeded two thirds of the slow-wave amplitudes per participant.

## Estimation of neural complexity: Higuchi Fractal Dimension

We quantified the degree of information processing in the brain during and following word presentation during sleep. To this aim, we computed Higuchi Fractal Dimension (HFD) as an EEG-based measure of 'complexity' of ongoing neural activity. HFD provides a non-linear measure of the complexity, variability, and randomness of EEG time-series (*Lau et al., 2022*; *Parbat and Chakraborty, 2021*). Importantly, a high neural complexity has been associated with successful memory formation during wakefulness (*Sheehan et al., 2018*). We computed HFD separately for each trial and electrode for the EEG time-series of our windows of interest (see below) using the MATLAB code provided by *Monge-Alvarez, 2024*. $K_{max}$ for extracting HFD was set to 10 (*Monge-Alvarez, 2024*). Single-trial HFD values were averaged within participants separately for each condition (experimental condition and control condition), electrode, and time window of interest. Average HFD values were then used for group-level analyses.

We determined the degree of stimulus-induced information processing following word offset. To this aim, we computed HFD for the time window from 0.5 to 2 s following word onset. We used the HFD values, which were computed for the time window from –2 to –0.5 s before word onset, as a baseline in order to control for inter-individual differences in neural complexity. To test for statistical differences in HFD, we performed mass univariate analyses with cluster-based permutation statistics (1000 permutations) to correct for multiple comparisons (*Maris and Oostenveld, 2007*).

## Acknowledgements

This work was supported by the Interfaculty Research Cooperation "Decoding Sleep: From Neurons to Health & Mind" of the University of Bern. We thank the undergraduate students Benjamin Ambühl, Sophie Ankner, Esther Brill, Samantha Glatt, Elena Grebenarov, Ronja Imlig, Leona Knüsel, Laura Schmid and Nicole Skieresz for their help with the data collection. Moreover, we thank Marc Züst and Marina Wunderlin for generating and validating the stimulus material.

## Additional information

### Funding

| Funder | Grant reference number | Author |
|---|---|---|
| University of Bern | Interfaculty Research Cooperation Grant "Decoding Sleep: From Neurons to Health & Mind" | Katharina Henke |
| Schweizerischer Nationalfonds zur Förderung der Wissenschaftlichen Forschung | SNSF Advanced Grant | Katharina Henke |
| Schweizerischer Nationalfonds zur Förderung der Wissenschaftlichen Forschung | TMAG-1_209374 | Katharina Henke |

The funders had no role in study design, data collection and interpretation, or the decision to submit the work for publication.

### Author contributions

Flavio J Schmidig, Conceptualization, Data curation, Software, Formal analysis, Investigation, Visualization, Methodology, Writing – original draft; Simon Ruch, Conceptualization, Data curation, Software, Formal analysis, Supervision, Validation, Writing - review and editing; Katharina Henke, Conceptualization, Supervision, Funding acquisition, Methodology, Project administration, Writing - review and editing

## Author ORCIDs
Flavio J Schmidig (iD) https://orcid.org/0000-0001-5257-0038
Simon Ruch (iD) http://orcid.org/0000-0002-5796-4543
Katharina Henke (iD) http://orcid.org/0000-0002-7895-044X

## Ethics
Informed consent was obtained from all participants. The study protocol was approved by the local ethics committee 'Kantonale Ethikkommission Bern number 2017-01046'.

Reviewer #1 (Public Review): https://doi.org/10.7554/eLife.89601.3.sa1
Reviewer #3 (Public Review): https://doi.org/10.7554/eLife.89601.3.sa2
Author response https://doi.org/10.7554/eLife.89601.3.sa3

---

# Additional files

## Supplementary files
• MDAR checklist

## Data availability
Raw files of the behavioral and neuronal data are available at the Open Science Framework (OSF) repository (https://osf.io/mafw6/). Code for the closed-loop targeting of slow-wave phases can be found in Ruch et al. (2022) or at the OSF repository (https://osf.io/ecvq8/). Further requests should be directed to and will be fulfilled by the corresponding author Flavio J. Schmidig.

The following dataset was generated:

| Author(s) | Year | Dataset title | Dataset URL | Database and Identifier |
|---|---|---|---|---|
| Schmidig F, Ruch S, Henke K | 2024 | Episodic long-term memory formation during slow-wave sleep | https://osf.io/mafw6/ | Open Science Framework, mafw6 |

The following previously published dataset was used:

| Author(s) | Year | Dataset title | Dataset URL | Database and Identifier |
|---|---|---|---|---|
| Ruch S, Schmidig F, Knüsel L, Henke K | 2022 | Closed-loop modulation of local slow oscillations in human NREM sleep | https://osf.io/ecvq8 | Open Science Framework, ecvq8 |

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
