## [Editor Report · eLife assessment]

This manuscript supports the intriguing idea that some aspects of novel learning can occur during sleep and outside of awareness. The authors provide **solid** evidence that presenting participants with novel words and their translations during sleep, especially during slow oscillation troughs, leads to the ability to categorize the semantic meaning of those words during awake testing 36 hours later. These findings represent a **valuable** contribution to the literature on unconscious processing and learning during sleep, although the claim that the results reflect episodic memory formation, in particular, deviates from the typical use of this term in the literature.

---

## [Referee Report · Reviewer #1 (Public Review)]

The authors show that concurrently presenting foreign words and their translations during sleep leads to the ability to semantically categorize the foreign words above chance. Specifically, this procedure was successful when stimuli were delivered during slow oscillation troughs as opposed to peaks, which has been the focus of many recent investigations into the learning & memory functions of sleep. Finally, further analyses showed that larger and more prototypical slow oscillation troughs led to better categorization performance, which offers hints to others on how to improve or predict the efficacy of this intervention.

Comments on the revised version:

I applaud the authors on a nice rebuttal. Many responses use solid arguments based on the existing literature, such as their response regarding the possibility that low-level acoustic characteristics explaining EEG differences between conditions. Their new analyses also clarify the paper. Additionally, I appreciate their labeling their more speculative claims as such. Below are my remaining thoughts:

Major point:

The largest remaining issue for me regards the term 'episodic'. Before I begin, I should say that I imagine the authors have thought considerably about this definition and may disagree with what I will say. That would be fine - it's their choice at this journal. My main point in writing this is to help them clarify their case further. R3 had a similar concern on the first round of review, and I imagine others holding the "traditional" view of episodic memory would be similarly skeptical. If the authors have a great rebuttal to these points, I imagine it will address others' concerns too.

I believe I understand the authors' argument: I read the Henke (2010, Nature Reviews Neuroscience) piece years back with great interest and again now, and I've gone back to read their other papers cited in this manuscript. Again, I applaud the authors on producing a large collection of fascinating findings expanding knowledge of what can be accomplished via unconscious learning. That includes this paper! But I still disagree with the term 'episodic' for what is measured here. The authors state in the Methods section that they prompted participants to 'guess whether the presented pseudoword designates an animal, a tool, or a place'. IMHO, the main issue of using 'episodic' is the nature of the memory representation - 'guessing' does not ask participants anything about the source (the who-what-when-why-where) of the information (anything about an episode).

Notably, it does seem to fit their own definition from Henke (2010). Rapid? I believe so - 4 trial-learning is fairly quick. Certainly, there are studies of supposed episodic memory that use a few rounds of learning the same stimuli (rather than single trial learning) and one can still get away with calling the nature of the memories 'episodic'. Flexible? I believe the authors mean that their task is flexible because participants learn a category exemplar during sleep (e.g., 'aryl'-'bird') but then only respond based on its category membership ('animal'?). If this is the case, I agree that the representations are flexible. Reliant on the 'episodic memory system' (lines 495-9)? Reasonably likely, given their prior findings (e.g., Züst et al., 2019). However, there is considerable data suggesting the hippocampus contributes to functions beyond episodic memory, including statistical learning (e.g., Schapiro et al., 2013, Current Biology), motor learning (e.g., Schendan et al., 2003, Neuron; Dohring et al., 2017, Cortex; Jacobacci et al., 2020, PNAS), attention (e.g., Aly & Turk-Browne, 2016, Cerebral Cortex), perception (e.g., Lee et al., 2012), and semantic memory (e.g., Cutler et al., 2019, Frontiers in Human Neuroscience). Therefore, given that the hippocampus contributes to other tasks too, saying the task is episodic in part because it likely relies on the hippocampus (the 'episodic memory system') is an incorrect reverse inference. But regardless of this concern, it seems true to me that the term fits 'episodic' according to Henke (2010).

So, it seems I'm raising an issue with this entire way of defining memory. IMHO, the biggest issue is that there is no reason to assume the participant relies upon any source-related information in making their guess. There is room in the field for a new type of rapid, unconscious, flexible, hippocampal-dependent learning that does not need to align with the term, 'episodic', for it to be important and fascinating! The term, 'episodic', is convenient for a reason - namely, for labeling the behavioral output of what it measures, not the process that underlies it. The authors have continually made an excellent case for rapid, unconscious, flexible, hippocampal-dependent learning, and it would seem even more beneficial for the field for the authors to just call this its own thing.

A related point:

- I see that the authors do not use 'episodic' in prior papers with similar tasks (e.g., Züst et al., 2019), and I am curious if anything changed in their thinking or why they use the term now. They can ignore this if they'd like, but it would perhaps give useful context.

Other points:

IMHO, the issue of repeated tests is more legitimate than the authors suggest. They state in their response letter, "However, recent literature suggests that retrieval practice is only beneficial when corrective feedback is provided (Belardi et al., 2021; Metcalfe, 2017)." This is incorrect. While retrieval practice is often less effective without feedback, it can be effective without feedback if retrieval accuracy is high and if the experimenters later employ a long enough retention interval to witness long-term effects. This is clear in various papers (e.g., Roediger & Karpicke, 2006, Psychological Science; Karpicke & Roediger, 2008, Science) and there is a nice theoretical model explaining how these complex effects could arise (Halamish & Bjork, 2011, JEP:LMC; Kornell et al., 2011, JML). The authors do not heavily rely on this in their paper, but they could consider tempering their claims that it is 'unlikely' (line 509) that delayed retrieval was affected by the first retrieval.

The authors claim that fast spindles are part of a speculative model underlying their learning effects (lines 605-6). However, they did not find any differential spindle effects in determining later performance, so they could consider keeping just points #1&2 or mentioning that spindles differ by condition but may not directly influence the learning effects here.

---

## [Referee Report · Reviewer #3 (Public Review)]

This is a revision in response to the reviewer's comments. The authors provided new analyses and try to acknowledge limitations, overall doing a good job, but the interpretation still seems to me going above the available evidence, especially for the claim that it is episodic memory formation during sleep. I still believe the paper will be fairer in dropping this speculative part and omitting the word "episodic" from the title (like actually they did in the abstract). The argument of the authors is that they refer to a computational definition of episodic memory, which is to some extent valid, but I am afraid it is not the way it will be understood by most readers, and it will thus indirectly contribute to an erroneous (or at least, not substantiated) interpretation of the brain's sleeping capabilities.

My main concern is that I have not seen any proposal for a control condition allowing to exclude the alternative, simpler hypothesis that mere perceptual associations between two elements (foreign word and translation) have been created and stored during sleep (which, I repeat, is already in itself an interesting finding). The authors argue that it seems to them not an efficient processing, but this an opinion, not a demonstration.

---

## [Author Response]

The following is the authors’ response to the original reviews.

We thank the reviewers and the editors for their careful reading of our manuscript and for the detailed and constructive feedback on our work. Please find attached the revised version of the manuscript. We performed an extensive revision of the manuscript to address the issues raised by the referees. We provide new analyses (regarding the response consistency and the neural complexity), added supplementary figures and edits to figures and texts. Based on the reviewers’ comments, we introduced several major changes to the manuscript.

Most notably, we

added a limitation statement to emphasize the speculative nature of our interpretation of the timing of word processing/associative bindingemphasized the limitations of the control conditionadded analyses on the interaction between memory retrieval after 12h versus 36hclarified our definition of episodic memoryadded detailed analyses of the “Feeling of having heard” responses and the confidence ratings

We hope that the revised manuscript addresses the reviewers' comments to their satisfaction. We believe that the revised manuscript has been significantly improved owing to the feedback provided. Below you can find a point-by-point response to each reviewer comment in blue. We are looking forward that the revision will be published in the Journal eLife.

**Reviewer #1 (Public Review):**
The authors show that concurrently presenting foreign words and their translations during sleep leads to the ability to semantically categorize the foreign words above chance. Specifically, this procedure was successful when stimuli were delivered during slow oscillation troughs as opposed to peaks, which has been the focus of many recent investigations into the learning & memory functions of sleep. Finally, further analyses showed that larger and more prototypical slow oscillation troughs led to better categorization performance, which offers hints to others on how to improve or predict the efficacy of this intervention. The strength here is the novel behavioral finding and supporting physiological analyses, whereas the biggest weakness is the interpretation of the peak vs. trough effect.R1.1. Major importance:I believe the authors could attempt to address this question: What do the authors believe is the largest implication of this studies? How far can this technique be pushed, and how can it practically augment real-world learning?

We revised the discussion to put more emphasis on possible practical applications of this study (lines 645-656).

In our opinion, the strength of this paper is its contribution to the basic understanding of information processing during deep sleep, rather than its insights on how to augment realworld learning. Given the currently limited data on learning during sleep, we believe it would be premature to make strong claims about potential practical applications of sleep-learning. In addition, as pointed out in the discussion section, we do not know what adverse effects sleep-learning has on other sleep-related mechanisms such as memory consolidation.

R1.2. Lines 155-7: How do the authors argue that the words fit well within the half-waves when the sounds lasted 540 ms and didn't necessarily start right at the beginning of each half-wave? This is a major point that should be discussed, as part of the down-state sound continues into the up-state. Looking at Figure 3A, it is clear that stimulus presented in the slow oscillation trough ends at a time that is solidly into the upstate, and would not neurolinguists argue that a lot of sound processing occurs after the end of the sound? It's not a problem for their findings, which is about when is the best time to start such a stimulus, but it's a problem for the interpretation. Additionally, the authors could include some discussion on whether possibly presenting shorter sounds would help to resolve the ambiguities here.

The word pairs’ presentations lasted on average ~540 ms. Importantly, the word pairs’ onset was timed to occur 100 ms before the maximal amplitude of the targeted peaks/troughs.

Therefore, most of a word’s sound pattern appeared during the negative going half-wave (about 350ms of 540ms). Importantly, Brodbeck and colleagues (2022) have shown that phonemes are continuously analyzed and interpreted with delays of about 50-200 ms, peaking at 100ms delay. These results suggest that word processing started just following the negative maximum of a trough and finished during the next peak. Our interpretation (e.g. line 520+) suggests that low-level auditory processing reaches the auditory cortex before the positive going half-wave. During the positive going half-wave the higher-level semantic networks appear the extract the presented word's meaning and associate the two simultaneously presented words. We clarified the time course regarding slow-wave phases and sound presentation in the manuscript (lines 158-164). Moreover, we added the limitation that we cannot know for sure when and in which slow-wave phase words were processed (lines 645-656). Future studies might want to look at shorter lasting stimuli to narrow down the timing of the word processing steps in relation to the sleep slow waves.

R1.3. Medium importance:Throughout the paper, another concern relates to the term 'closed-loop'. It appears this term has been largely misused in the literature, and I believe the more appropriate term here is 'real-time' (Bergmann, 2018, Frontiers in Psychology; Antony et al., 2022, Journal of Sleep Research). For instance, if there were some sort of algorithm that assessed whether each individual word was successfully processed by the brain during sleep and then the delivery of words was subsequently changed, that could be more accurately labelled as 'closed-loop'.

We acknowledge that the meaning of “closed-loop” in its narrowest sense is not fulfilled here. We believe that “slow oscillation phase-targeted, brain-state-dependent stimulation” is the most appropriate term to describe the applied procedure (BSDBS, Bergmann, 2018). We changed the wording in the manuscript to brain-state-dependent stimulation algorithm. Nevertheless, we would like to point out that the algorithm we developed and used (TOPOSO) is very similar to the algorithms often termed closed-loop algorithm in memory and sleep (e.g. Esfahani et al., 2023; Garcia-Molina et al., 2018; Ngo et al., 2013, for a comparison of TOPOSO to these techniques see Wunderlin et al., 2022 and for more information about TOPOSO see Ruch et al., 2022).

R1.4. Figure 5 and corresponding analyses: Note that the two conditions end up with different sounds with likely different auditory complexities. That is, one word vs. two words simultaneously likely differ on some low-level acoustic characteristics, which could explain the physiological differences. Either the authors should address this via auditory analyses or it should be added as a limitation.

This is correct, the two conditions differ on auditory complexities. Accordingly, we added this issue as another limitation of the study (line 651-653). We had decided for a single word control condition to ensure that no associative learning (between pseudowords) could take place in the control condition because this was the critical learning process in the experimental condition. We would like to point out that we observed significant differences in brain responses to the presentation of word-pairs (experimental condition) vs single pseudowords (control condition) in the Trough condition, but not the Peak condition. If indeed low-level acoustic characteristics explained the EEG differences occurring between the two conditions then one would expect these differences occurring in both the trough and the peak condition because earlier studies showed that low-level acoustic processing proceeds in both phases of slow waves (Andrillon et al., 2016; Batterink et al., 2016; Daltrozzo et al., 2012).

R1.5. Line 562-7 (and elsewhere in the paper): "episodic" learning is referenced here and many times throughout the paper. But episodic learning is not what was enhanced here. Please be mindful of this wording, as it can be confusing otherwise.

The reported unconscious learning of novel verbal associations during sleep may not match textbook definitions of episodic memory. However, the traditional definitions of episodic memory have long been criticised (e.g., Dew & Cabeza, 2011; Hannula et al., 2023; Henke, 2010; Reder et al., 2009; Shohamy & Turk-Browne, 2013).

We stand by our claim that sleep-learning was of episodic nature. Here we use a computational definition of episodic memory (Cohen & Eichenbaum, 1993; Henke, 2010; O’Reilly et al., 2014; O’Reilly & Rudy, 2000) and not the traditional definition of episodic memory that ties episodic memory to wakefulness and conscious awareness (Gabrieli, 1998; Moscovitch, 2008; Schacter, 1998; Squire & Dede, 2015; Tulving, 2002). We revised the manuscript to clarify that and how our definition differs from traditional definitions. Please see reviewer comment R3.1 for a more extensive answer.

**Reviewer #2 (Public Review):**
In this project, Schmidig, Ruch and Henke examined whether word pairs that were presented during slow-wave sleep would leave a detectable memory trace 12 and 36 hours later. Such an effect was found, as participants showed a bias to categorize pseudowords according to a familiar word that they were paired with during slow-wave sleep. This behavior was not accompanied by any sign of conscious understanding of why the judgment was made, and so demonstrates that long-term memory can be formed even without conscious access to the presented content. Unconscious learning occurred when pairs were presented during troughs but not during peaks of slow-wave oscillations. Differences in brain responses to the two types of presentation schemes, and between word pairs that were later correctly- vs. incorrectly-judged, suggest a potential mechanism for how such deep-sleep learning can occur.The results are very interesting, and they are based on solid methods and analyses. Results largely support the authors' conclusions, but I felt that there were a few points in which conclusions were not entirely convincing:R2.1. As a control for the critical stimuli in this study, authors used a single pseudoword simultaneously played to both ears. This control condition (CC) differs from the experimental condition (EC) in a few dimensions, among them: amount of information provided, binaural coherence and word familiarity. These differences make it hard to conclude that the higher theta and spindle power observed for EC over CC trials indicate associative binding, as claimed in the paper. Alternative explanations can be made, for instance, that they reflect word recognition, as only EC contains familiar words.

We agree. In the revised version of the manuscript, we emphasise this as a limitation of our study (line 653-656). Moreover, we understand that the differences between stimuli of the control and the experimental condition must not rely only on the associative binding of two words. We cautioned our interpretation of the findings.

Interestingly, EC vs CC exhibits differences following trough- but not peak targeting (see R1.4). If indeed all the EC vs CC differences were unrelated to associative binding, we would expect the same EC vs CC differences when peaks were targeted. Hence, the selective EC vs CC differences in the trough condition suggest that the brain is more responsive to sound, information, word familiarity and word semantics during troughs, where we found successful learning, compared to peaks, where no learning occurred. Troughtargeted word pairs (EC) versus foreign words (CC) enhanced the theta power 336 at 500 ms following word onset and this theta enhancement correlated significantly with interindividual retrieval performance indicating that theta probably promoted associative learning during sleep. This correlation was insignificant for spindle power.

R2.2. The entire set of EC pairs were tested both following 12 hours and following 36 hours. Exposure to the pairs during test #1 can be expected to have an effect over memory one day later, during test #2, and so differences between the tests could be at least partially driven by the additional activation and rehearsal of the material during test #1. Therefore, it is hard to draw conclusions regarding automatic memory reorganization between 12 and 36 hours after unconscious learning. Specifically, a claim is made regarding a third wave of plasticity, but we cannot be certain that the improvement found in the 36 hour test would have happened without test #1.

We understand that the retrieval test at 12h may have had an impact on performance on the retrieval test at 36h. Practicing retrieval of newly formed memories is known to facilitate future retrieval of the same memories (e.g. Karpicke & Roediger, 2008). Hence, practicing the retrieval of sleep-formed memories during the retrieval test at 12h may have boosted performance at 36h.

However, recent literature suggests that retrieval practice is only beneficial when corrective feedback is provided (Belardi et al., 2021; Metcalfe, 2017). In our study, we only presented the sleep-played pseudowords at test and participants received no feedback regarding the accuracy of their responses. Thus, a proper conscious re-encoding could not take place. Nevertheless, the retrieval at 12h may have altered performance at 36h in other ways. For example, it could have tagged the reactivated sleep-formed memories for enhanced consolidation during the next night (Rabinovich Orlandi et al., 2020; Wilhelm et al., 2011).

We included a paragraph on the potential carry-over effects from retrieval at 12h on retrieval at 36h in the discussion section (line 489-496; line 657-659). Furthermore, we removed the arguments about the “third wave of plasticity”.

R2.3. Authors claim that perceptual and conceptual processing during sleep led to increased neural complexity in troughs. However, neural complexity was not found to differ between EC and CC, nor between remembered and forgotten pairs. It is therefore not clear to me why the increased complexity that was found in troughs should be attributed to perceptual and conceptual word processing, as CC contains meaningless vowels. Moreover, from the evidence presented in this work at least, I am not sure there is room to infer causation - that the increase in HFD is driven by the stimuli - as there is no control analysis looking at HFD during troughs that did not contain stimulation.

With the analysis of the HFD we would like to provide an additional perspective to the oscillation-based analysis. We checked whether the boundary condition of Peak and Trough targeting changes the overall complexity or information content in the EEG. Our goal was to assess the change in neural complexity (relative to a pre-stimulus baseline) following the successful vs unsuccessful encoding of word pairs during sleep.

We acknowledge that a causal interpretation about HFD is not warranted, and we revised the manuscript accordingly. It was unexpected that we could not find the same results in the contrast of EC vs CC or correct vs incorrect word pairs. We suggest that our signal-to noise ratio might have been too weak.

One could argue that the phase targeting alone (without stimulation) induces peak/trough differences in complexity. We cannot completely rule out this concern. But we tried to use the EEG that was not influenced by the ongoing slow-wave: the EEG 2000-500ms before the stimulus onset and 500-2000ms after the stimulus onset. Therefore, we excluded the 1s of the targeted slow-wave, hoping that most of the phase inherent complexity should have faded out (see Figure 2). We could not further extend the time window of analysis due to the minimal stimulus onset interval of 2s. Of course we cannot exclude that the targeted Trough impacted the following HFD. We clarified this in the manuscript (line 384-425).

Furthermore, we did find a difference of neural complexity between the pre-stimulus baseline and the post-stimulus complexity in the Peak condition but not in the Trough condition (we now added this contrast to the manuscript, line 416-419). Hence, the change in neural complexity is a reaction to the interaction of the specific slow-wave phase with the processing of the word pairs. Even though these results cannot provide unambiguous, causal links, we think they can figure as an important start for other studies to decipher neural complexity during slow wave sleep.

**Reviewer #3 (Public Review):**
The study aims at creating novel episodic memories during slow wave sleep, that can be transferred in the awake state. To do so, participants were simultaneously presented during sleep both foreign words and their arbitrary translations in their language (one word in each ear), or as a control condition only the foreign word alone, binaurally. Stimuli were presented either at the trough or the peak of the slow oscillation using a closed-loop stimulation algorithm. To test for the creation of a flexible association during sleep, participant were then presented at wake with the foreign words alone and had (1) to decide whether they had the feeling of having heard that word before, (2) to attribute this word to one out of three possible conceptual categories (to which translations word actually belong), and (3) to rate their confidence about their decision.R3.1. The paper is well written, the protocol ingenious and the methods are robust. However, the results do not really add conceptually to a prior publication of this group showing the possibility to associate in slow wave sleep pairs of words denoting large or small object and non words, and then asking during ensuing wakefulness participant to categorise these non words to a "large" or "small" category. In both cases, the main finding is that this type of association can be formed during slow wave sleep if presented at the trough (versus the peak) of the slow oscillation. Crucially, whether these associations truly represent episodic memory formation during sleep, as claimed by the authors, is highly disputable as there is no control condition allowing to exclude the alternative, simpler hypothesis that mere perceptual associations between two elements (foreign word and translation) have been created and stored during sleep (which is already in itself an interesting finding). In this latter case, it would be only during the awake state when the foreign word is presented that its presentation would implicitly recall the associated translation, which in turn would "ignite" the associative/semantic association process eventually leading to the observed categorisation bias (i.e., foreign words tending to be put in the same conceptual category than their associated translation). In the absence of a dis-confirmation of this alternative and more economical hypothesis, and if we follow Ocam's razor assumption, the claim that there is episodic memory formation during sleep is speculative and unsupported, which is a serious limitation irrespective of the merits of the study. The title and interpretations should be toned down in this respect

Our study conceptually adds to and extends the findings by Züst et al. (a) by highlighting the precise time-window or brain state during which sleep-learning is possible (e.g. slow-wave trough targeting), (b) by demonstrating the feasibility of associative learning during night sleep, and (c) by uncovering the longevity of sleep-formed memories.

We acknowledge that the reported unconscious learning of novel verbal associations during sleep may not match textbook definitions of episodic memory. However, the traditional definitions of episodic memory have long been criticised e.g, (Dew & Cabeza, 2011; Hannula et al., 2023; Henke, 2010; Reder et al., 2009; Shohamy & Turk-Browne, 2013). We stand by our claim that sleep-learning was of episodic nature. We use a computational definition of episodic memory (Cohen & Eichenbaum, 1993; Henke, 2010; O’Reilly et al., 2014; O’Reilly & Rudy, 2000), and not the traditional definition of episodic memory that ties episodic memory to wakefulness and conscious awareness (Gabrieli, 1998; Moscovitch, 2008; Schacter, 1998; Squire & Dede, 2015; Tulving, 2002). The core computational features of episodic memory are (1) rapid learning, (2) association formation, and (3) a compositional and flexible representation of the associations in long-term memory.

Therefore, we revised the manuscript to emphasize how our definition differs from traditional definitions (line 64).

For the current study, we designed a retrieval task that calls on the core computational features of episodic memory by assessing flexible retrieval of sleep-formed compositional word-word associations. Reviewer 3 suggests an alternative interpretation for the learning observed here: mere perceptual associations between foreign words and translations words are stored during sleep, and semantic associations are only inferred at retrieval testing during ensuing wakefulness. First, these processing steps would require the rapid soundsound associative encoding, long-term storage, and the flexible sound retrieval, which would still require hippocampal processing and computations in the episodic memory system. Second, this mechanism seems highly laborious and inefficient. The sound pattern of a word at 12 hours after learning triggers the reactivation of an associated sound pattern of another word. This sound pattern then elicits the activation of the translation words’ semantics leading to the selection of the correct superordinate semantic category at test.

Overall, we believe that our pairwise-associative learning paradigm triggered a rapid conceptual-associative encoding process mediated by the hippocampus that provided for flexible representations of foreign and translation words in episodic memory. This study adds to the existing literature by examining specific boundary conditions of sleep-learning and demonstrates the longevity (at least 36 hours) of sleep-learned associations.

Other remarks:

R3.2. Lines 43-45 : the assumption that the sleeping brain decides whether external events can be disregarded, requires awakening or should be stored for further consideration in the waking state is dubious, and the supporting references date from a time (the 60') during which hypnopedia was investigated in badly controlled sleep conditions (leaving open the doubt about the possibility that it occurred during micro awakenings)

We revised the manuscript to add timelier and better controlled studies that bolster the 60ties-born claim (line 40-51). Recently, it has been shown that the sleeping brain preferentially processes relevant information. For example the information conveyed by unfamiliar voices (Ameen et al., 2022), emotional content (Holeckova et al., 2006; Moyne et al., 2022), our own compared to others’ names (Blume et al., 2018).

R3.3. 1st paragraph, lines 48-53 , the authors should be more specific about what kind of new associations and at which level they can be stored during sleep according to recent reports, as a wide variety of associations (mostly elementary levels) are shown in the cited references. Limitations in information processing during sleep should also be acknowledged.

In the lines to which R3 refers, we cite an article (Ruch & Henke, 2020) in which two of the three authors of the current manuscript elaborate in detail what kind of associations can be stored during sleep. We revised these lines to more clearly present the current understanding of the potential and the limitations of sleep-learning (line 40-51). Although information processing during sleep is generally reduced (Andrillon et al., 2016), a variety of different kinds of associations can be stored, ranging from tone-odour to word-word association (Arzi et al., 2012, 2014; Koroma et al., 2022; Züst et al., 2019).

R3.4. The authors ran their main behavioural analyses on delayed retrieval at 36h rather than 12h with the argument that retrieval performance was numerically larger at 36 than 12h but the difference was non-significant (line 181-183), and that effects were essentially similar. Looking at Figure 2, is the trough effect really significant at 12h ? In any case, the fact that it is (numerically) higher at 36 than 12h might suggest that the association created at the first 12h retrieval (considering the alternative hypothesis proposed above) has been reinforced by subsequent sleep.

The Trough effect at 12h is not significant, as stated on line 185 (“Planned contrasts against chance level revealed that retrieval performance significantly exceeded chance at 36 hours only (P36hours = 0.036, P12hours = 0.094).”). It seems that our wording was not clear. Therefore, we refined the description of the behavioural analysis in the manuscript (lines 188-193).

In brief, we report an omnibus ANOVA with a significant main effect of targeting type (Trough vs Peak, main effect Peak versus Trough: F(1,28) = 5.237, p = 0.030, d = 0.865). Because Trough-targeting led to significantly better memory retention than Peak-targeting, we computed a second ANOVA, solely including participants with through-targeted word-pair encoding. The memory retention in the Trough condition is above chance (MTrough = 39.11%, SD = 10.76; FIntercept (1,14) = 5.660, p = 0.032) and does not significantly differ between the 12h and 36h retrieval (FEncoding-Test Delay (1,14) = 1.308, p = 0.272). However, the retrieval performance at 36h numerically exceeds the performance at 12h and the direct comparison against chance reveals that the 36h but not the 12h retrieval was significant (P36hours = 0.036, P12hours = 0.094). Hence, we found no evidence for above chance performance at the 12h retrieval and focused on the retrieval after 36h in the EEG analysis.

We agree with the reviewer that the subsequent sleep seems to have improved consolidation and subsequent retrieval. We assume that the reviewer suggests that participants merely formed perceptual associations during sleep and encoded episodic-like associations during testing at 12h (as pointed out in R 3.1). However, we believe that it is unlikely that the awake encoding of semantic associations during the 12h retrieval led to improved performance after 36h. We changed the discussion regarding the interaction between retrieval at 12h and 36h (line 505-512, also see R 2.2)

R3.5> In the discussion section lines 419-427, the argument is somehow circular in claiming episodic memory mechanisms based on functional neuroanatomical elements that are not tested here, and the supporting studies conducted during sleep were in a different setting (e.g. TMR)

Indeed, the TMR and animal studies are a different setting compared to the present study. We re-wrote this part and only focused on the findings of Züst and colleagues (2019), who examined hippocampal activity during the awake retrieval of sleep-formed memories (lines 472-482). Additionally, we would like to emphasise that our main reasoning is that the task requirements called upon the episodic memory system.

R3.6. Supplementary Material: in the EEG data the differentiation between correct and incorrect ulterior classifications when presented at the peak of the slow oscillation is only significant in association with 36h delayed retrieval but not at 12h, how do the authors explain this lack of effect at 12 hour ?

We assume that the reviewer refers to the TROUGH condition (word-pairs targeted at a slow-wave trough) and not as written to the peak condition. We argue that the retention performance at 12h is not significantly above chance (M12hours = 37.4%, P12hours = 0.094).

Hence, the distinction between “correctly” and “incorrectly” categorised word pairs was not informative for the EEG analysis during sleep. For whatever reason the 12h retrieval was not significantly above chance, the less successful memory recall and thus a less balanced trial count makes recall accuracy a worse delineator for separating EEG trials then the recall performance after 36 hours.

**Recommendations for the authors:**

**Reviewer #1 (Recommendations For The Authors):**
Minor importance:Abstract: The opening framing is confusing here and in the introduction. Why frame the paper in the broadest terms about awakenings and threats from the environment when this is a paper about intersections between learning & memory and sleep? I do understand that there is an interesting point to be made about the counterintuitive behavioral findings with respect to sleep generally being perceived as a time when stimuli are blocked out, but this does not seem to me to be the broadest points or the way to start the paper. The authors should consider this but of course push back if they disagree.

We understand the reviewer’s criticism but believe that this has more to do with personal preferences than with the scientific value or validity of our work. We believe that it is our duty as researchers to present our study in a broader context because this may help readers from various fields to understand why the work is relevant. To some readers, evidence for learning during sleep may seem trivial, to others, it may seem impossible or a weird but useless conundrum. By pointing out potential evolutionary benefits of the ability to acquire new information during sleep, we help the broad readership of eLife understand the relevance of this work.

Lines 31-32: "Neural complexity" -> "neural measures of complexity" because it isn't clear what "neural complexity" means at this point in the abstract. Though, note my other point that I believe this analysis should be removed.

To our understanding, “neural complexity” is a frequently used term in the field and yields more than 4000 entries on google scholar. Whereas ‘neural measures of complexity’ only finds 3 hits on google scholar [September 2023]. In order to link our study with other studies on neural complexity, we would like to keep this terminology. As an example, two recent publications using “neural complexity” are Lee et al. (2020) and Frohlich et al. (2022).

Lines 42-43: The line of work on 'sentinel' modes would be good to cite here (e.g., Blume et al., 2017, Brain & Language).

We added the suggested citation to the manuscript (lines 52).

Lines 84-90: While I appreciate the authors desire to dig deep and try to piece this all together, this is far too speculative in my opinion. Please see my other points on the same topic.

In this paragraph, we point out why both peaks and troughs are worth exploring for their contributions to sensory processing and learning during sleep. Peaks and troughs are contributing mutually to sleep-learning. Our speculations should inspire further work aimed at pinning down the benefits of peaks and troughs for sleep-learning. We clarified the purpose and speculative nature of our arguments in the revised version of the manuscript.

Line 109: "outlasting" -> "lasting over" or "lasting >"

We changed the wording accordingly.

Line 111: I believe 'nonsense' is not the correct term here, and 'foreign' (again) would be preferred. Some may be offended to hear their foreign word regarded as 'nonsense'. However, please let me know if I have misunderstood.

We would like to use the linguistic term “pseudoword” (aligned with reviewer 2’s comment) and we revised the manuscript accordingly.

Figure 1A: "Enconding" -> "Encoding"

Thank you for pointing this out.

Lines 201-2: Were there interactions between confidence and correctness on the semantic categorization task? Were correct responses given with more confidence than incorrect ones? This would not necessarily be a problem for the authors' account, as there can of course be implicit influences on confidence (i.e., fluency).

As is stated in the results section, confidence ratings did not differ significantly between correct and incorrect assignments (Trough condition: F(1,14) = 2.36, p = 0.15); Peak condition: F(1,14) = 0.48, p = 0.50.

Line 236: "Nicknazar" -> "Niknazar"

Thank you for pointing this out.

Line 266: "profited" -> "benefited"

We changed the wording accordingly.

Lines 280-4: There seems some relevance here with Malerba et al. (2018) and her other papers to categorize slow oscillations.

Diving into the details on how to best categorise slow oscillations is beyond the scope of this manuscript. Here, we build on work from the field of microstate analyses and use two measures to describe and quantify the targeted brain states: the topography of the electric field (i.e., the correlation of the electric field with an established template or “microstate”), and the field strength (global field power, GFP). While the topography of a quasi-stable electric field reflects activity in a specific neural network, the strength (GFP) of a field most likely mirrors the degree of activation (or inactivity) in the specific network. Here, we find that consistent targeting of a specific network state yielding a strong frontal negativity benefitted learning during sleep. For a more detailed explanation of the slow-wave phase targeting see (Ruch et al., 2022).

Lines 343-6: Was it intentional to have 0.5 s (0.2-0.7 s) surrounding the analysis around 500 ms but only 0.4 s (0.8-1.2 s) surrounding the analysis around 1 s? Could the authors use the same size interval or justify having them be different?

We apologise for the misleading phrasing and we clarified this in the revised manuscript. We applied the same procedure for the comparison of later correctly vs incorrectly classified pseudowords as we did for the comparison between EC and CC. Hence, we analysed the entire window from 0s to 2.5s with a cluster-based permutation approach. Contrary to the EC vs CC contrast, no cluster remained significant for the comparison of the subsequent memory effect. By mistake we reported the wrong time window. In the revised manuscript, the paragraph is corrected (lines 364-369).

Line 356-entire HFD section: it is unclear what's gained by this analysis, as it could simply be another reflection of the state of the brain at the time of word presentation. In my opinion, the authors should remove this analysis and section, as it does not add clarity to other aspects of the paper.(If the authors keep the section) Line 361-2 - "Moreover, high HFD values have been associated with cognitive processing (Lau et al., 2021; Parbat & Chakraborty, 2021)." This statement is vague. Could the authors elaborate?

Please see our answer to Reviewer 2 (2.3) for a more detailed explanation. In brief, we would like to keep the analysis with the broad time window of -2 to -0.5 and from 0.5 to 2 s.

Lines 403-4: How was it determined that these neural networks mediated bothconscious/unconscious processes? Perhaps the authors meant to make a different point, but the way it reads to me is that there is evidence that some neural networks are conscious and others are not and both forms engage in similar functions.

We revised the manuscript to be more precise and clear: “The conscious and unconscious rapid encoding and flexible retrieval of novel relational memories was found to recruit the same or similar networks including the hippocampus(Henke et al., 2003; Schneider et al., 2021). This suggests that conscious and unconscious relational memories are processed by the same memory system.” (p. 22, top).

Lines 433-41: Performance didn't actually significantly increase from 12 to 36 hours, so this is all too speculative in my opinion.

We removed the speculative claim that performance may have increased from the retrieval at 12 hours to the retrieval at 36 hours.

Line 534: "assisted by enhanced" -> "coincident with". It's unclear whether theta reflects successful processing as having occurred or whether it directly affects or assists with it.

We have adjusted the wording to be more cautious, as suggested (line 588).

Line 572-4: Rothschild et al. (2016) is relevant here.

Unfortunately, we do not see the relevance of this article within the context of our work.

Line 577 paragraph: The authors may consider adding a note on the importance of ethical considerations surrounding this form of 'inception'.

We extended this part by adding ethical considerations to the discussion section (Stickgold et al., 2021, line 657).

Line 1366: It would be better if the authors could eventually make their data publicly available. This is obviously not required, but I encourage the authors to consider it if they have not considered it already.In my opinion, the discussion is too long. I really appreciate the authors trying to figure out the set of precise times in which each level of neural processing might occur and how this intersects with their slow oscillation phase results. However, I found a lot of this too speculative, especially given that the sounds may bleed into parts of other phases of the slow oscillation. I do not believe this is a problem unique to these authors, as many investigators attempting to target certain phases in the target memory reactivation literature have faced the same problem, but I do believe the authors get ahead of the data here. In particular, there seems to be one paragraph in the discussion that is multiple pages long (p. 22-24). This paragraph I believe has too much detail and should be broken up regardless, as it is difficult for the reader to follow.

Considering the recent literature, we believe this interpretation best explains the data. As argued earlier, we believe that a speculative interpretation of the reported phenomena can provide substantial added value because it inspires future experimental work. We have improved the manuscript by clearly distinguishing between data and interpretation. We do declare the speculative nature of some offered interpretations. We hope that these speculations, which are testable hypotheses (!), will eventually be confirmed or refuted experimentally.

**Reviewer #2 (Recommendations For The Authors):**
I very much enjoyed the paper and think it describes important findings. I have a few suggestions for improvement, and minor comments that caught my eye during reading:(1) I was missing an analysis of CC ERP, and its comparison to EC ERP.

We added this analysis to the manuscript (line 299-301). The comparison of CC ERP with EC ERP did not yield any significant cluster for either the peak (cluster-level Monte Carlo p=0.54) or the trough (cluster-level Monte Carlo p>0.37). We assume that the noise level was too high for the identification of differences between CC and EC ERP.

(2) Regarding my public review comment #2, some light can be shed on between-test effects, I believe, using an item-based analysis - looking at correlations between items' classifications in test #1 and test #2. The assumption seems to be that items that were correct in test #1 remained correct in test #2 while other new correct classifications were added, owing to the additional consolidation happening between the two tests. But that is an empirical question that can be easily tested. If no consistency in item classification is found, on the other hand, or if only consistency in correct classification is found, that would be interesting in itself. This item-based analysis can help tease away real memory from random correct classification. For instance, the subset of items that are consistently classified correctly could be regarded as non-fluke at higher confidence and used as the focus of subsequent-memory analysis instead of the ones that were correct only in test #2.

Thanks, we re-analysed the data accordingly. Participants were consistent at choosing a specific object category for an item at 12 hours and 36 hours (consistency rate = 47% same category, chance level is 1/3). Moreover, the consistency rate did not differ between the Trough and the Peak condition (MTrough = 47.2%, MPeak = 47.0%, P = 0.98). The better retrieval performance in the Trough compared to the Peak condition after 36 hours is due to: (A) if participants were correct at 12h, they chose again the correct answer at 36h (Trough: 20% & Peak: 14%). (B) Following an incorrect answer at 12h, participants switched to another object category at 36h (Trough: 72%, Peak: 67%). (C) If participants switched the object category following an incorrect answer at 12h, they switched more often to the correct category at 36h in the trough versus the peak condition (Trough: in 56% & Peak: 53%).Hence, the data support the reviewer’s assumption: items that were correct after 12 hours remained correct after 36 hours, while other new correct classifications were generated at 36h owing to the additional consolidation happening between the two tests. We added this finding to the manuscript (line 191-200, Figure S6):

As suggested, we re-analysed the ERP with respect to the subsequent memory effect. This time we computed four conditions according to the reviewer’s argument about consistently correctly classified pseudowords, presented in the figure below: ERP of trials that were correctly classified at 36h (blue), ERP of trials that were incorrectly classified at 36h (light blue), ERP of trials that were correctly classified twice (brown) and ERP of trials that were not correctly classified twice (orange, all trials that are not in brown). Please note that the two blue lines are reported in the manuscript and include all trials. The brown and the orange line take the consistency into account and together include as well all trials.

**Author response image 2. sa3fig2:** 

By excluding even more trials from the group of correct retrieval responses, the noise level gets high. Therefore, the difference between the twice-correct and the not-twice-correct trials is not significant (cluster-level Monte Carlo p > 0.27). Because the ERP of twice-correct trials seems very similar to the ERP of the trials correctly classified at 36h at frontal electrodes, we assume that our ERP effect is not driven by a few extreme subjects. Similarly, not-twicecorrect trials (orange) have a stronger frontal trough than the trials incorrectly classified at 36h (light blue).

(3) In a similar vein, a subject-based analysis would be highly interesting. First and foremost, readers would benefit from seeing the lines that connect individual dots across the two tests in figures 2B and 2C. It is reasonable to expect that only a subset of participants were successful learners in this experiment. Finding them and analyzing their results separately could be revealing.

We added a Figure S1 to the supplementary material, providing the pairing between performance of the 12h and the 36h retrieval.

It is an interesting idea to look at successful learners alone. We computed the ERP of the subsequent memory effect for those participants, who had an above change retrieval accuracy at 36h. The result shows a similar effect as reported for all participants (frontal cluster ~0-0.3s). The p-value is only 0.08 because only 9 of 15 participants exhibited an above chance retrieval performance at 36 hours.

**Author response image 3. sa3fig3:** 

ERP effect of correct (blue) vs incorrect (light blue) pseudoword category assignment of participants with a retrieval performance above chance at 36h (SD as shades):

We prefer to not include this data in the manuscript, but are happy to provide it here.

(4) I wondered why the authors informed subjects of the task in advance (that they will be presented associations when they slept)? I imagine this may boost learning as compared to completely naïve subjects. Whether this is the reason or not, I think an explanation of why this was done is warranted, and a statement whether authors believe the manipulation would work otherwise. Also, the reader is left wondering why subjects were informed only about test #1 and not about test #2 (and when were they told about test #2).

Subjects were informed of all the tests upfront. We apologize for the inconsistency in the manuscript and revised the method part. The explanation of why participants were informed is twofold: (a) Participants had to sleep with in-ear headphones. We wanted to explain to participants why these are necessary and why they should not remove them. (b) We hoped that participants would be expecting unconsciously sounds played during sleep, would process these sounds efficiently and would remain deeply asleep (no arousals).

(5) FoHH is a binary yes/no question, and so may not have been sensitive enough to demonstrate small differences in familiarity. For comparison, the Perceptual Awareness Scale (Ramsøy & Overgaard, 2004) that is typically used in studies of unconscious processing is of a 4-point scale, and this allows to capture more nuanced effects such as partial consciousness and larger response biases. Regardless, it would be informative to have the FoHH numbers obtained in this study, and not just their comparison between conditions. Also, was familiarity of EC and CC pseudowords compared? One may wonder whether hearing the pseudowords clearly vs. in one ear alongside a familiar word would make the word slightly more familiar.

We apologize for having simplified this part too much in the manuscript. Indeed, the FoHH is comparable to the PAS. We used a 4-point scale, where participants rated their feeling of whether they have heard the pseudoword during previous sleep. In the revised manuscript, we report the complete results (line 203-223). The FoHH did not differ between any of the suggested contrasts. Thus, for both the peak and the trough condition, the FoHH did not differ between sleep-played vs new; correct EC trials vs new; correct vs incorrect EC trials; EC vs CC trials. To illustrate the results, a figure of the FoHH has been added to the supplement (Figure S4).

(6) Similarly, it would be good to report the numbers of the confidence ratings in the paper as well.

In the revised manuscript, we extended the description of the confidence rating results. We added the descriptive statistics (line 224-236) and included a corresponding figure in the supplement (Figure S5).

Minor/aesthetic comments:

We implemented all the following suggestions.

(1) I suggest using "pseudoword" or "nonsense word" instead of "foreign word", because "foreign word" typically means a real word from a different language. It is quite confusing when starting to read the paper.

After reconsidering, we think that pseudoword is the appropriate linguistic term and have revised the manuscript accordingly.

(2) Lines 1000-1001: "The required sample size of N = 30 was determined based on a previous sleep-learning study". I was missing a description of what study you are referring to.(3) I am not sure I understood the claim nor the rationale made in lines 414-417. Is the claim that pairs did not form one integrated engram? How do we know that? And why would having one engram not enable extracting the meaning from a visual-auditory presentation of the cue? The sentence needs some rewording and/or unpacking.(4) Were categories counterbalanced (i.e., did each subjects' EC contain 9 animal words, 9 tool words and 9 place words)?(5) Asterisks indicating significant effects are missing from Figure 4 and S2.(6) Fig1 legend: "Participants were played with pairs" is ungrammatical.(7) Line 1093: no need for a comma.(8) Line 1336: missing opening parenthesis(9) Line 430: "observe" instead of "observed".(10) Line 466: two dots instead of one..
**Reviewer #3 (Recommendations For The Authors):**
Methods: 2 separate ANOVAs are performed (lines 160-185), but would not it make more sense to combine both in one ? If kept separated then a correction for multiple comparisons might be needed (p/2 = 0.025)

We computed an omnibus ANOVA. In a next step, we examined the effect in the significant targeting condition by computing another ANOVA. For further explanations, see reviewer comment 3.4.

References

Ameen, M. S., Heib, D. P. J., Blume, C., & Schabus, M. (2022). The Brain Selectively Tunes to Unfamiliar Voices during Sleep. Journal of Neuroscience, 42(9), 1791–1803. https://doi.org/10.1523/JNEUROSCI.2524-20.2021

Andrillon, T., Poulsen, A. T., Hansen, L. K., Léger, D., & Kouider, S. (2016). Neural Markers of Responsiveness to the Environment in Human Sleep. The Journal of Neuroscience, 36(24), Article 24. https://doi.org/10.1523/JNEUROSCI.0902-16.2016

Arzi, A., Holtzman, Y., Samnon, P., Eshel, N., Harel, E., & Sobel, N. (2014). Olfactory Aversive Conditioning during Sleep Reduces Cigarette-Smoking Behavior. Journal of Neuroscience, 34(46), Article 46. https://doi.org/10.1523/JNEUROSCI.2291-14.2014

Arzi, A., Shedlesky, L., Ben-Shaul, M., Nasser, K., Oksenberg, A., Hairston, I. S., & Sobel, N. (2012). Humans can learn new information during sleep. Nature Neuroscience, 15(10), Article 10. https://doi.org/10.1038/nn.3193

Batterink, L. J., Creery, J. D., & Paller, K. A. (2016). Phase of Spontaneous Slow Oscillations during Sleep Influences Memory-Related Processing of Auditory Cues. Journal of Neuroscience, 36(4), 1401–1409. https://doi.org/10.1523/JNEUROSCI.3175-15.2016

Belardi, A., Pedrett, S., Rothen, N., & Reber, T. P. (2021). Spacing, Feedback, and Testing Boost Vocabulary Learning in a Web Application. Frontiers in Psychology, 12. https://www.frontiersin.org/articles/10.3389/fpsyg.2021.757262

Bergmann, T. O. (2018). Brain State-Dependent Brain Stimulation. Frontiers in Psychology, 9, 2108. https://doi.org/10.3389/fpsyg.2018.02108

Blume, C., del Giudice, R., Wislowska, M., Heib, D. P. J., & Schabus, M. (2018). Standing sentinel during human sleep: Continued evaluation of environmental stimuli in the absence of consciousness. NeuroImage, 178, 638–648. https://doi.org/10.1016/j.neuroimage.2018.05.056

Brodbeck, C., & Simon, J. Z. (2022). Cortical tracking of voice pitch in the presence of multiple speakers depends on selective attention. Frontiers in Neuroscience, 16. https://www.frontiersin.org/articles/10.3389/fnins.2022.828546

Cohen, N. J., & Eichenbaum, H. (1993). Memory, Amnesia, and the Hippocampal System. A Bradford Book.

Daltrozzo, J., Claude, L., Tillmann, B., Bastuji, H., & Perrin, F. (2012). Working memory is partially preserved during sleep. PloS One, 7(12), Article 12.

Dew, I. T. Z., & Cabeza, R. (2011). The porous boundaries between explicit and implicit memory: Behavioral and neural evidence. Annals of the New York Academy of Sciences, 1224(1), 174–190. https://doi.org/10.1111/j.1749-6632.2010.05946.x

Esfahani, M. J., Farboud, S., Ngo, H.-V. V., Schneider, J., Weber, F. D., Talamini, L. M., & Dresler, M. (2023). Closed-loop auditory stimulation of sleep slow oscillations: Basic principles and best practices. Neuroscience & Biobehavioral Reviews, 153, 105379. https://doi.org/10.1016/j.neubiorev.2023.105379

Frohlich, J., Chiang, J. N., Mediano, P. A. M., Nespeca, M., Saravanapandian, V., Toker, D., Dell’Italia, J., Hipp, J. F., Jeste, S. S., Chu, C. J., Bird, L. M., & Monti, M. M. (2022). Neural complexity is a common denominator of human consciousness across diverse regimes of cortical dynamics. Communications Biology, 5(1), Article 1. https://doi.org/10.1038/s42003-022-04331-7

Gabrieli, J. D. E. (1998). Cognitive neuroscience of human memory. Annual Review of Psychology, 87–115.

Garcia-Molina, G., Tsoneva, T., Jasko, J., Steele, B., Aquino, A., Baher, K., Pastoor, S., Pfundtner, S., Ostrowski, L., Miller, B., Papas, N., Riedner, B., Tononi, G., & White, D. P. (2018). Closed-loop system to enhance slow-wave activity. Journal of Neural Engineering, 15(6), 066018. https://doi.org/10.1088/1741-2552/aae18f

Hannula, D. E., Minor, G. N., & Slabbekoorn, D. (2023). Conscious awareness and memory systems in the brain. WIREs Cognitive Science, 14(5), e1648. https://doi.org/10.1002/wcs.1648

Henke, K. (2010). A model for memory systems based on processing modes rather than consciousness. Nature Reviews Neuroscience, 11(7), Article 7. https://doi.org/10.1038/nrn2850

Henke, K., Mondadori, C. R. A., Treyer, V., Nitsch, R. M., Buck, A., & Hock, C. (2003). Nonconscious formation and reactivation of semantic associations by way of the medial temporal lobe. Neuropsychologia, 41(8), Article 8. https://doi.org/10.1016/S0028-3932(03)00035-6

Holeckova, I., Fischer, C., Giard, M.-H., Delpuech, C., & Morlet, D. (2006). Brain responses to a subject’s own name uttered by a familiar voice. Brain Research, 1082(1), 142–152. https://doi.org/10.1016/j.brainres.2006.01.089

Karpicke, J. D., & Roediger, H. L. (2008). The Critical Importance of Retrieval for Learning. Science, 319(5865), 966–968. https://doi.org/10.1126/science.1152408

Koroma, M., Elbaz, M., Léger, D., & Kouider, S. (2022). Learning New Vocabulary Implicitly During Sleep Transfers With Cross-Modal Generalization Into Wakefulness. Frontiers in Neuroscience, 16, 801666. https://doi.org/10.3389/fnins.2022.801666

Lee, Y., Lee, J., Hwang, S. J., Yang, E., & Choi, S. (2020). Neural Complexity Measures. Advances in Neural Information Processing Systems, 33, 9713–9724. https://proceedings.neurips.cc/paper/2020/hash/6e17a5fd135fcaf4b49f2860c2474c7 c-Abstract.html

Metcalfe, J. (2017). Learning from Errors. Annual Review of Psychology, 68(1), 465–489. https://doi.org/10.1146/annurev-psych-010416-044022

Moscovitch, M. (2008). The hippocampus as a “stupid,” domain-specific module: Implications for theories of recent and remote memory, and of imagination. Canadian Journal of Experimental Psychology/Revue Canadienne de Psychologie Expérimentale, 62, 62–79. https://doi.org/10.1037/1196-1961.62.1.62

Moyne, M., Legendre, G., Arnal, L., Kumar, S., Sterpenich, V., Seeck, M., Grandjean, D., Schwartz, S., Vuilleumier, P., & Domínguez-Borràs, J. (2022). Brain reactivity to emotion persists in NREM sleep and is associated with individual dream recall. Cerebral Cortex Communications, 3(1), tgac003. https://doi.org/10.1093/texcom/tgac003

Ngo, H.-V. V., Martinetz, T., Born, J., & Mölle, M. (2013). Auditory Closed-Loop Stimulation of the Sleep Slow Oscillation Enhances Memory. Neuron, 78(3), Article 3. https://doi.org/10.1016/j.neuron.2013.03.006

O’Reilly, R. C., Bhattacharyya, R., Howard, M. D., & Ketz, N. (2014). Complementary Learning Systems. Cognitive Science, 38(6), 1229–1248. https://doi.org/10.1111/j.1551-6709.2011.01214.x

O’Reilly, R. C., & Rudy, J. W. (2000). Computational principles of learning in the neocortex and hippocampus. Hippocampus, 10(4), 389–397. https://doi.org/10.1002/1098-1063(2000)10:4<389::AID-HIPO5>3.0.CO;2-P

Rabinovich Orlandi, I., Fullio, C. L., Schroeder, M. N., Giurfa, M., Ballarini, F., & Moncada, D. (2020). Behavioral tagging underlies memory reconsolidation. Proceedings of the National Academy of Sciences, 117(30), 18029–18036. https://doi.org/10.1073/pnas.2009517117

Reder, L. M., Park, H., & Kieffaber, P. D. (2009). Memory systems do not divide on consciousness: Reinterpreting memory in terms of activation and binding. Psychological Bulletin, 135(1), Article 1. https://doi.org/10.1037/a0013974

Ruch, S., & Henke, K. (2020). Learning During Sleep: A Dream Comes True? Trends in Cognitive Sciences, 24(3), 170–172. https://doi.org/10.1016/j.tics.2019.12.007

Ruch, S., Schmidig, F. J., Knüsel, L., & Henke, K. (2022). Closed-loop modulation of local slow oscillations in human NREM sleep. NeuroImage, 264, 119682. https://doi.org/10.1016/j.neuroimage.2022.119682

Schacter, D. L. (1998). Memory and Awareness. Science, 280(5360), 59–60. https://doi.org/10.1126/science.280.5360.59

Schneider, E., Züst, M. A., Wuethrich, S., Schmidig, F., Klöppel, S., Wiest, R., Ruch, S., & Henke, K. (2021). Larger capacity for unconscious versus conscious episodic memory. Current Biology, 31(16), 3551-3563.e9. https://doi.org/10.1016/j.cub.2021.06.012

Shohamy, D., & Turk-Browne, N. B. (2013). Mechanisms for widespread hippocampal involvement in cognition. Journal of Experimental Psychology: General, 142(4), 1159–1170. https://doi.org/10.1037/a0034461

Squire, L. R., & Dede, A. J. O. (2015). Conscious and Unconscious Memory Systems. Cold Spring Harbor Perspectives in Biology, 7(3), a021667. https://doi.org/10.1101/cshperspect.a021667

Stickgold, R., Zadra, A., & Haar, A. J. H. (2021). Advertising in Dreams is Coming: Now What? Dream Engineering. https://dxe.pubpub.org/pub/dreamadvertising/release/1

Tulving, E. (2002). Episodic Memory: From Mind to Brain. Annual Review of Psychology, 53(1), 1–25. https://doi.org/10.1146/annurev.psych.53.100901.135114

Wilhelm, I., Diekelmann, S., Molzow, I., Ayoub, A., Mölle, M., & Born, J. (2011). Sleep Selectively Enhances Memory Expected to Be of Future Relevance. Journal of Neuroscience, 31(5), 1563–1569. https://doi.org/10.1523/JNEUROSCI.3575-10.2011

Wunderlin, M., Koenig, T., Zeller, C., Nissen, C., & Züst, M. A. (2022). Automatized online prediction of slow-wave peaks during non-rapid eye movement sleep in young and old individuals: Why we should not always rely on amplitude thresholds. Journal of Sleep Research, 31(6), e13584. https://doi.org/10.1111/jsr.13584

Züst, M. A., Ruch, S., Wiest, R., & Henke, K. (2019). Implicit Vocabulary Learning during Sleep Is Bound to Slow-Wave Peaks. Current Biology, 29(4), 541-553.e7. https://doi.org/10.1016/j.cub.2018.12.038